# JudgeLM: Fine-tuned Large Language Models are Scalable Judges

Lianghui Zhu[1,2] *        Xinggang Wang[1†]        Xinlong Wang[2†]

[1] School of EIC, Huazhong University of Science & Technology
[2] Beijing Academy of Artificial Intelligence

Code & Models: https://github.com/baaivision/JudgeLM

## Abstract

Evaluating Large Language Models (LLMs) in open-ended scenarios is challenging because existing benchmarks and metrics can not measure them comprehensively. To address this problem, we propose to fine-tune L**LM**s as scalable **judge**s (JudgeLM) to evaluate LLMs efficiently and effectively in open-ended benchmarks. We first propose a comprehensive, large-scale, high-quality dataset containing task seeds, LLMs-generated answers, and GPT-4-generated judgments for fine-tuning high-performance judges, as well as a new benchmark for evaluating the judges. We train JudgeLM at different scales from 7B, 13B, to 33B parameters, and conduct a systematic analysis of its capabilities and behaviors. We then analyze the key biases in fine-tuning LLM as a judge and consider them as position bias, knowledge bias, and format bias. To address these issues, JudgeLM introduces a bag of techniques including swap augmentation, reference support, and reference drop, which clearly enhance the judge's performance. JudgeLM obtains the state-of-the-art judge performance on both the existing PandaLM benchmark and our proposed new benchmark. Our JudgeLM is efficient and the JudgeLM-7B only needs 3 minutes to judge 5K samples with 8 A100 GPUs. JudgeLM obtains high agreement with the teacher judge, achieving an agreement exceeding 90% that even surpasses human-to-human agreement[1]. JudgeLM also demonstrates extended capabilities in being judges of the single answer, multimodal models, multiple answers, multi-turn chat, etc.

## 1 Introduction

Recent advancements in large language models (LLMs) have fostered significant interest due to their remarkable performance in following instructions and their broad capabilities in dealing with open-ended scenarios. Based on the open-source LLMs, including OPT (Zhang et al., 2022), Flan-T5 (Chung et al., 2022), LLaMA (Touvron et al., 2023a), and Pythia (Biderman et al., 2023), researchers propose numerous methods to align these models with human preferences through instruction fine-tuning. These aligned LLMs demonstrate enhanced abilities in comprehending human instructions and generating more coherent responses. Nonetheless, existing benchmarks (Hendrycks et al., 2020; Liang et al., 2022) and traditional metrics (Lin, 2004; Papineni et al., 2002; Zhang et al., 2019; Sellam et al., 2020; Yuan et al., 2021) do not adequately estimate the capabilities of LLMs in open-ended scenarios. Therefore, a new benchmark method that could evaluate LLMs comprehensively in open-ended tasks is needed.

Concurrent works are making efforts to explore various methods for evaluating the performance of LLM. The arena-format (Zheng et al., 2023) methods leverage crowdsourced platforms to extract anonymous LLM competition results. While evaluations by humans are trustworthy, they are also time-consuming and financially demanding. Some approaches (Chiang et al., 2023) utilize GPT-4 as

---

*This work was done when Lianghui Zhu was an intern at Beijing Academy of Artificial Intelligence.
†Corresponding authors: xgwang@hust.edu.cn and wangxinlong@baai.ac.cn.

[1]As a reference, the max agreement among humans in MT-bench (Zheng et al., 2023) is 82%.

a judge. Nevertheless, these methods grapple with challenges of potential data exposure and volatile API model transitions, potentially compromising the judge's reproducibility. PandaLM (Wang et al., 2023) attempts to fine-tune open-source LLMs for evaluating answers. However, limitations stemming from the training data quality, and inherent LLM biases, undermine the effectiveness of such fine-tuned models in the role of a judge.

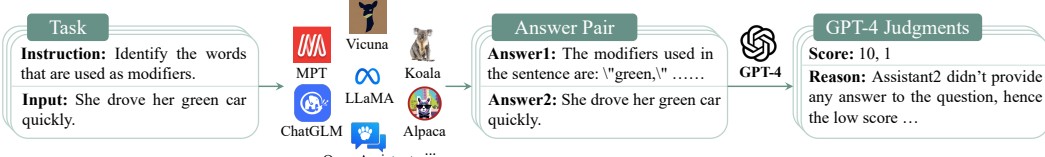

(a) Data generation pipeline of our JudgeLM. We first collect 105K seed tasks as questions. Then, we extract answers from 11 LLMs and randomly sample a pair of answers from the answer set. Last, we input the tasks, the sampled answer pairs, and optionally reference answers to GPT-4, which generates scores and detailed reasons as a judge teacher.

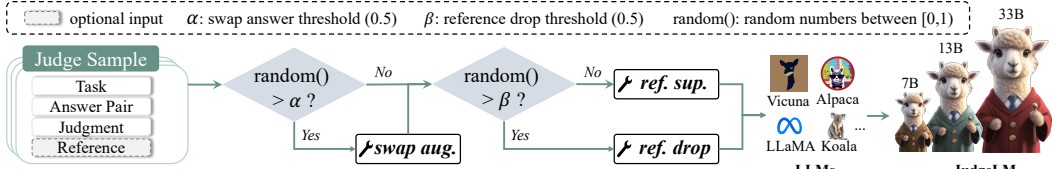

(b) An illustration of the JudgeLM's fine-tuning and various functions. We use generated judge samples to fine-tune LLMs as scalable judges. When fine-tuning LLMs as judges, we also propose swap augmentation, reference support, and reference drop to address the position bias, knowledge bias, and format bias, respectively.

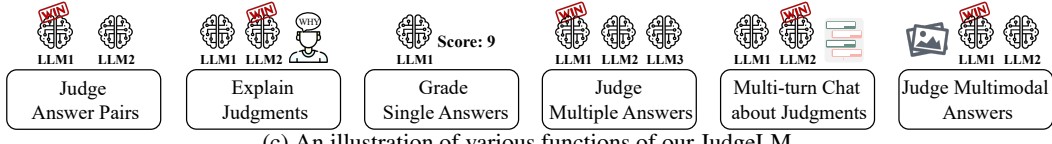

(c) An illustration of various functions of our JudgeLM.

Figure 1: An overview of our scalable JudgeLM including data generation, fine-tuning, and various functions.

In this paper, we propose to evaluate LLMs through fine-tuned open-source **LLM**s, which serve as scalable **judge**s (JudgeLM) achieving satisfactory agreement with the teacher judge. Our methodology incorporates scalable judges as evaluators in open-ended tasks, coupled with a high-quality dataset conducive to both training and evaluating the judge models. Within our framework, we adapt open-source LLMs to serve as judges and analyze their scaling ability in relation to model size (ranging from 7B to 33B) and volume of training data (extending from 3.5K to 100K). Our curated dataset comprises 105K seed questions, LLM answer pairs, and judgments from the teacher judge, GPT-4, as shown in Fig. 1a. Note that we generated two judgments for each seed task with and without reference answers. This dataset is partitioned, with 100K seed questions allocated for training (2 × larger than PandaLM) and the remainder for validation (29 × larger than PandaLM).

Utilizing LLMs as judges inevitably introduces biases such as position bias (favoring answers in specific positions), knowledge bias (over-reliance on pre-trained knowledge), and format bias (optimal performance only under specific prompt formats) as shown in Fig. 8, 10, 12, 13. When fine-tuning is not possible, GPT-4-API-based judge (Zheng et al., 2023) tries to alleviate this by well-designed prompt methods, i.e., Chain-of-thought, few-shot judge, and judging multiple times with different positions. JudgeLM presents a new way that can address these biases in the fine-tuning stage, skipping the complicated prompt methods and multi-turn API calling. Moreover, our JudgeLM system presents extended capabilities as shown in Fig. 1b, including grading single answers, judging multiple answers, judging multimodal models, multi-turn chat, etc.

In contrast to arena-format methods, our approach is rapid and has a low cost. For instance, JudgeLM-7B requires only 8 A100 GPUs and can evaluate 5000 response pairs in just 3 minutes. In comparison to closed-source LLM judges, JudgeLM ensures reproducibility and protects user privacy. When compared to concurrent open-source LLM judges, our system explores both the scaling ability and biases in LLM fine-tuning. Furthermore, JudgeLM dataset stands as the most diverse and high-quality one, significantly benefitting subsequent research in judge model investigations.

Our main contributions can be summarized as follows:

- We introduce a high-quality, large-scale dataset for judge models, enriched with diverse seed tasks, LLMs-generated answers, and detailed judgments from GPT-4, laying the foundation for future LLMs evaluating research.

- We propose JudgeLM, a scalable language model judge, designed for evaluating LLMs in open-ended scenarios. It achieves an agreement exceeding 90% that surpasses the human-to-human agreement. Our JudgeLM can also generalize to many extended tasks.

- We analyze the biases inherent to **LLM judge fine-tuning** and introduce a series of methods to address them. Our methods significantly improve the consistency of the model in different cases, making the JudgeLM more reliable and flexible.

## 2 RELATED WORK

### 2.1 INSTRUCTION FINE-TUNING OF LARGE LANGUAGE MODELS

With the development of large language models (LLMs), researchers find that fine-tuning pre-trained LLMs such as GPT-3 (Brown et al., 2020), T5 (Raffel et al., 2020), OPT (Zhang et al., 2022), and PaLM (Chowdhery et al., 2022) enable LLMs to follow human instructions and help with open-ended tasks. The instruction fine-tuned LLMs such as InstructGPT (Ouyang et al., 2022), Chat-GPT (OpenAI, 2022), FLAN-T5 (Chung et al., 2022), FLAN-PaLM (Chung et al., 2022), OPT-IML (Iyer et al., 2022), and GPT-4 (OpenAI, 2023) exhibit stronger ability in zero-shot or few-shot tasks than their base models. After Meta released the powerful open-source LLM LLaMA (Touvron et al., 2023a) and LLaMA2 (Touvron et al., 2023b), lots of instruction fine-tuning works based on LLaMA or LLaMA2 were proposed in the natural language generation or multimodal generation domain, such as Alpaca, Vicuna (Chiang et al., 2023), OpenFlamingo (Awadalla et al., 2023), LLaMA-Adapter (Zhang et al., 2023), and Emu (Sun et al., 2023). Our JudgeLM also belongs to the LLaMA family and takes the Vicuna series as base models. Our JudgeLM follows the instruction fine-tuning manner to create LLM judges and proposes to model the judgment-generation task as "grading, judging, and reasoning". We further collect a high-quality, large-scale dataset for research in judging the performance of LLMs.

### 2.2 EVALUATION OF LARGE LANGUAGE MODELS

As many open-source large language models (LLMs) and their fine-tuned variants are proposed and present remarkable performance on various tasks, evaluating the capabilities of LLMs becomes a popular and challenging task. To address this problem, Chatbot Arena (Zheng et al., 2023) aims to build a crowdsourced platform that ranks the LLMs through pairwise comparison and Elo rating. The crowdsourced way to evaluate LLMs has more reliable results but faces high costs and low efficiency. Vicuna (Chiang et al., 2023) uses GPT-4 as a judge to select the better answer. Although the GPT-4-based method can judge LLMs like a human expert, the API-based methods have potential risks of data leakage and unstable performance. Zeno Build (Alex & Graham, 2023) proposes to evaluate LLMs at a customer service dataset, but using traditional metrics such as ChrF (Popović, 2015) and BERTScore (Zhang et al., 2019) can not fully evaluate the answers of LLMs in open-ended tasks. Besides, PandaLM (Wang et al., 2023) and Auto-J Li et al. (2023a) developed judge models based on LLaMA (Touvron et al., 2023a) or LLaMA2 (Touvron et al., 2023b) to compare answers produced by LLMs. When serving as judges, PandaLM achieves an accuracy close to Chat-GPT but ignoring the inherent LLM biases limits its performance further. Our JudgeLM contains scalable judges from 7B-parameter to 33B-parameter and achieves state-of-the-art performance in both PandaLM and our benchmarks. Furthermore, researchers can use the proposed JudgeLM locally which ensures reproducibility and data security.

## 3 DATASET

High-quality, large-scale datasets are crucial for effectively fine-tuning large language models (LLMs) to act as evaluative judges. However, the concurrent datasets, such as the one by PandaLM (Wang et al., 2023), present limitations in terms of diversity and the granularity of judgment

---

**Input**

**Question**
If Rosslyn Mountain Boys had produced two more albums, how many albums would they have produced?Rosslyn Mountain Boys was a band that backed up a number of nationally known singers, and released two albums of their own.

**Answer 1**
4

**Answer 2**
Rosslyn Mountain Boys produced

**Ground Truth**
Rosslyn Mountain Boys produced two albums on their own, so if they produced two more albums then they would have produced four albums.

**Traditional Metrics**

| Answer | Rouge-1↑ | Rouge-2↑ | Rouge-L↑ | BLEU↑ | BERTScore↑ | BLEURT↑ | BARTScore↑ |
|--------|----------|----------|----------|-------|-----------|---------|-----------|
| 1 | 0.00 | 0.00 | 0.00 | 0.00 | 0.62 | -1.39 | -5.33 |
| 2 | 0.30 | 0.24 | 0.30 | 0.52 | 0.76 | -0.78 | -3.76 |

**Our Judgment Output**

**Answer 1's Score: 10**         **Answer 2's Score: 1**
Assistant 2's response is incomplete and does not provide any information or answer to the question, hence the low score. On the other hand, Assistant 1's response is accurate and directly answers the question. The Rosslyn Mountain Boys produced two albums, and if they had produced two more, they would have produced a total of four albums. Therefore, Assistant 1 receives a perfect score.

Figure 2: The input and output of our JudgeLM data sample. In open-ended scenarios, traditional metrics can not judge answers accurately by comparing the answers with ground truth. However, the LLM judges can understand the questions and answers and give accurate scores and reasons.

criteria. To address this, we introduce a novel dataset replete with a rich variety of seed tasks, comprehensive answers from modern LLMs, answers' grades from the teacher judge, and detailed reasons for judgments. Section 3.1 elucidates the data generation process, while Section 3.2 delineates the methods adopted for training and evaluation using our dataset.

## 3.1 DATA GENERATION

The primary objective of our data generation is to create a large-scale and diversified dataset that maximizes the evaluative capabilities of judge models. We sample 105K instruction seed tasks from a large-scale set that contains Alpaca-GPT4 (Peng et al., 2023), Dolly-15K (Conover et al., 2023), GPT4All-LAION (Anand et al., 2023), and ShareGPT. To enhance the heterogeneity of the dataset, answers are collated from 11 leading open-source LLMs including, but not limited to, LLaMA (Touvron et al., 2023a), Alpaca, and Vicuna (Chiang et al., 2023). Following this, we amalgamate LLM-generated answers with the reference answer to create answer sets. Pairs are randomly selected from the sets, upon which, fine-grained scores and detailed reasons are assigned by the advanced teacher model, GPT-4. To ensure robust and comprehensive judgments, we utilize detailed templates as demonstrated in Fig. 3. Additionally, to allow the model to judge with reference answers, the reference-inclusive template is employed as Fig. 4. This encourages the model to integrate external knowledge during the evaluative process. Please note that all samples in the JudgeLM *val* set are further checked and re-annotated by authors to ensure alignment with human preference.

## 3.2 TRAINING AND EVALUATING

To better utilize our dataset to train and evaluate the judge models, we partition it into a training split and a validation split. The training set contains 100K judge samples, while the validation set has 5K. We then introduce the way we use this dataset to train and evaluate, respectively.

**Training.** The training process of JudgeLM adheres to the instruction fine-tuning paradigm. As illustrated in Fig. 2, the model is fed a question alongside a pair of answers, and an optional reference answer, yielding outputs comprising scores and detailed reasons. It is imperative to note the significance of a detailed crafted prompt template to harness the full potential of JudgeLM's instruction-following ability. Distinct input templates cater to scenarios with and without references, as depicted in Fig. 3 and Fig. 4 respectively.

To further analyze the scaling ability of JudgeLM, we fine-tune JudgeLM with sizes of 7B, 13B, and 33B parameters. The specific hyperparameters are enumerated in Table 11. As for the scaling analysis for dataset size, we also fine-tune JudgeLM on varying data scales from 3.5K to 100K samples. JudgeLM demonstrates scaling ability both in terms of model size and data volume.

**Evaluating.** For the judge's result, we model it as "grading, judging, and reasoning". The judge model first generates scores for answer pairs. Subsequently, we can get the judge result from three situations: "Answer 1 wins" if the answer 1's score is higher than the answer 2's, "Answer 2 wins" if the answer 2's score is higher, or "Tie" if the scores of two answers are the same. Last, the model generates detailed reasons if needed. The advantage of this modeling is that the judge model just needs little time to grade and judge, and generates time-consuming reasoning optionally.

For the metrics, we employ the objective metrics and reliability metrics to evaluate the judge models comprehensively. For the objective metrics, we compute the agreement, precision, recall, and F1-score between the model's judge results and those of the teacher. This provides insights into the alignment of judge models with established benchmarks, such as GPT-4 or human experts. As for reliability metrics, we first compare the results before and after swapping LLM answers. Then we calculate the self-consistency to measure the judge model's reliability. Last, we further calculate the metrics like "bias toward 1st", "bias toward 2nd", and "delta bias" to get insights from specific position biases and their variance.

## 4    INHERENT BIAS

In this paper, we also study the inherent biases that influence the reliability of fine-tuned LLM judges through reliability metrics and visualizations.

**Position Bias.** Position bias means that the LLM judges prefer answers in a certain position and it widely exists in natural language processing tasks (Ko et al., 2020; Wang et al., 2018) and decision-making of humans (Blunch, 1984; Raghubir & Valenzuela, 2006). The powerful LLMs, ChatGPT and GPT-4, also face this challenge when working as judges (Wang et al., 2023; Zheng et al., 2023; Li et al., 2023b). As the qualitative and quantitative results shown in Fig. 8 and Table 5, JudgeLM also faces the position bias and prefers the first answer when swapping the positions of answers.

**Knowledge Bias.** Knowledge bias arises when the pre-trained data lacks the knowledge of some seed tasks or induces possibly undesirable knowledge (Ko et al., 2020; Zheng et al., 2023) that could degenerate the generative capabilities of LLMs. Fig. 10 provides an example that LLM judges can not give correct judgments to open-ended tasks if they lack related truth.

**Format Bias.** Researchers expect that the judge model can make judgments based on pre-trained knowledge when the reference is not available and can make judgments following the reference when it is available. However, our experiments revealed that judge models fine-tuned without reference perform poorly in judging with reference, and vice versa, as shown in Fig. 12, Fig. 13, and Table 6. We hypothesize fine-tuning with references encourages the judge model to make judgments based on external knowledge and fine-tuning without references pushes the judge model to make judgments through its pre-trained knowledge. We name the situation that a judge fine-tuned without reference but validated with reference as a mismatched format, and vice versa. Such a format bias limits the further generalization of the judge model in other domains.

## 5    METHOD

In evaluating LLM-generated answers for a seed question, the LLM judge aims to determine the superior answer from a pair of candidates. Motivated by recent methods (Touvron et al., 2023a; Chiang et al., 2023; Ouyang et al., 2022), we present JudgeLM, a scalable judge model, and address

inherent biases in such models. Our methodology is depicted in Fig. 1b. The subsequent sections provide a detailed breakdown of our approach.

## 5.1 SWAP AUGMENTATION

MT-bench (Zheng et al., 2023) and PandaLM (Wang et al., 2023) alleviate the position bias by judging twice with original and reverse order. These methods regard the result as a tie if the judgments are not the same. This kind of method ignoring the inherent position bias and casting double time to evaluate, can be regarded as a compromise and does not improve the reliability of LLM judges.

Intuitively, swapping the positions at the fine-tuning stage could push the judge model to pay more attention to the contents of answers rather than positions. Leveraging our structured judge data, we can easily swap the positions of answers to generate a new input sample. Correspondingly, we also swap the scores and question indexes of the judgment from the teacher (i.e., GPT4) to get the new ground truth. As shown in Fig. 15, the augmented judge sample keeps the same results but exchanges the positions of answers. Overall, it is simple but effective to augment the training data and address position bias. The JudgeLM-with-swap-augmentation can give good judgment to the same judge sample as shown in Fig. 9.

## 5.2 REFERENCE SUPPORT

Introducing external knowledge in the fine-tuning stage is an intuitive way to make up for the lack of related pre-trained knowledge. To do so, we propose the reference support method to teach the model to judge with the help of reference answers. Following Zheng et al. (2023), we collect reference answers for all judge samples and re-generate reference-guided judgments by GPT-4. Please note that GPT-4 also gives different scores and judgments for most judge samples with or without references. This proves that the differences between pre-trained knowledge and reference answers greatly impact judgments. As shown in Fig. 11, the JudgeLM with reference support can avoid factual errors and give reliable judgments. Furthermore, introducing reference support to LLM judges can simply insert judge preferences. JudgeLM with reference support training can flexibly set reference answers with different preferences for different scenarios and needs. As shown in Fig. 16, changing reference answers does not need extra training and makes JudgeLM more flexible to different preferences.

## 5.3 REFERENCE DROP

To address the format bias, we introduce a method, named reference drop, in which we randomly drop the training sample with reference and use the corresponding sample without reference. As shown in Fig. 14, judge models with reference drop can alleviate the overfitting for fine-tuning formats and make judgments based on external reference or pre-trained knowledge when given reference or not, respectively. Furthermore, the reference drop method also makes the judge model easy to use and decreases the cost of fitting into different formats.

## 6 EXPERIMENT

We study the performance of JudgeLM as follows: Section 6.1 presents the main results of JudgeLM comparing with concurrent methods, Section 6.2 analyzes the scaling ability of JudgeLM from both model sizes and data scales, and Section 6.3 shows ablation studies of proposed methods in detail. Detailed settings are shown in Section A.2.

## 6.1 MAIN RESULTS

**Comparison on JudgeLM Benchmark.** We first evaluate the proposed JudgeLM on our $val$ set. Note that JudgeLM $val$ set is further checked and re-annotated by authors to ensure alignment with human preference. As shown in Table 1, we give the quantitative results of GPT-3.5, Vicuna-13B, PandaLM-7B, Auto-J-13B (Li et al., 2023a), InstructScore-7B (Xu et al., 2023), and our JudgeLM with three model sizes. Among them, GPT-3.5 is used in the form of APIs with the help of templates in Fig. 3 and Fig. 4. PandaLM-7B and Auto-J-13B are deployed with the released checkpoints and

Table 1: Main results for our JudgeLM and concurrent methods on our *val* set, which uses GPT-4 annotation results as ground truth.

| Methods | Agreement ↑ (w/ GPT-4) | Precision ↑ (w/ GPT-4) | Recall ↑ (w/ GPT-4) | F1 ↑ (w/ GPT-4) | Consistency ↑ (w/ swap.) |
|---|---|---|---|---|---|
| ***Judge w/o reference.*** | | | | | |
| GPT-3.5 | 73.83 | 70.70 | 52.80 | 52.85 | 68.89 |
| Vicuna-13B | - | - | - | - | - |
| PandaLM-7B | 68.61 | 40.75 | 38.82 | 39.41 | 74.78 |
| Auto-J-13B | 74.86 | 61.65 | 57.53 | 58.14 | 84.34 |
| ***Judge w/o reference (Ours).*** | | | | | |
| JudgeLM-7B | 81.11 | 69.67 | 78.39 | 72.21 | 83.57 |
| JudgeLM-13B | 84.33 | 73.69 | 80.51 | 76.17 | 85.01 |
| JudgeLM-33B | 89.03 | 80.97 | 84.76 | 82.64 | 91.36 |
| ***Judge w/ reference.*** | | | | | |
| GPT-3.5 | 71.46 | 56.86 | 51.12 | 51.14 | 62.94 |
| Vicuna-13B | - | - | - | - | - |
| PandaLM-7B | 63.77 | 39.79 | 34.82 | 35.18 | 55.39 |
| Auto-J-13B | 72.90 | 58.80 | 56.12 | 56.59 | 82.84 |
| InstructScore-7B | 55.80 | 58.74 | 56.84 | 53.72 | - |
| ***Judge w/ reference (Ours).*** | | | | | |
| JudgeLM-7B | 84.08 | 75.92 | 82.55 | 78.28 | 84.46 |
| JudgeLM-13B | 85.47 | 77.71 | 82.90 | 79.77 | 87.23 |
| JudgeLM-33B | 89.32 | 84.00 | 86.21 | 84.98 | 92.37 |

Table 2: JudgeLM zero-shot evaluation results on PandaLM *test* set, which uses human annotation results as ground truth. "*" means the results are reported in PandaLM (Wang et al., 2023)

| Methods | Agreement ↑ (w/ Human) | Precision ↑ (w/ Human) | Recall ↑ (w/ Human) | F1 ↑ (w/ Human) |
|---|---|---|---|---|
| ***zero-shot methods.*** | | | | |
| GPT-3.5* | 62.96 | 61.95 | 63.59 | 58.20 |
| GPT-4* | 66.47 | 66.20 | 68.15 | 61.80 |
| ***Fine-tuned on PandaLM*** *train* **set.** | | | | |
| PandaLM-7B* | 59.26 | 57.28 | 59.23 | 54.56 |
| ***Ours (zero-shot).*** | | | | |
| JudgeLM-7B | 65.07 | 66.89 | 71.95 | 61.92 |
| JudgeLM-13B | 68.97 | 68.21 | 74.15 | 65.12 |
| JudgeLM-33B | 75.18 | 69.30 | 74.93 | 69.73 |

templates. These methods could be regarded as zero-shot methods because they are not fine-tuned by the JudgeLM dataset. On JudgeLM *val* set, the vanilla Vicuna-13B fails 77% of questions. Specifically, the vanilla Vicuna-13B can not even output a pair of scores in judgments in the failed cases. But the finetuned version, i.e., JudgeLM, would not fail any questions in the JudgeLM *val* set. Our JudgeLMs are fine-tuned with proposed methods, i.e., swap augmentation, reference support, and reference drop. So, they can handle situations with or without references simultaneously. It can be observed that our JudgeLM-7B outperforms PandaLM-7B, Auto-J, and InstructScore in all metrics, and even surpasses GPT-3.5. Furthermore, the proposed JudgeLM-33B exhibits the most powerful judge ability.

**Comparison on Other Human Evaluation Benchmarks.** We further evaluate our JudgeLM on other Human evaluation benchmarks, i.e., PandaLM *test* set and human-annotated MM-Vet. PandaLM's *train* and *val* sets are annotated by GPT-3.5 and humans, respectively. Following the manner of the PandaLM *val* set, we present the zero-shot results of JudgeLM in Table 2. It can be observed that the JudgeLM-7B outperforms GPT-3.5 and PandaLM-7B. When compared with GPT-4, JudgeLM-7B has lower accuracy and higher Precision, Recall, and F1-score than GPT-4. Furthermore, JudgeLM-33B achieves higher results than GPT-4, which demonstrates that fine-

Table 3: Efficiency comparison for our JudgeLM and PandaLM on our *val* set. We use a machine with 8 Nvidia-A100 GPUs with 40G memory to evaluate their efficiency.

| Methods | model size | GPUs per model | parallel judge? | generate reason? | total time |
|---|---|---|---|---|---|
| PandaLM | 7B | 1 | ✗ | ✓ | 6 hrs 40 mins |
| ***Ours.*** | | | | | |
| JudgeLM | 7B | 1 | ✗ | ✓ | 6 hrs 40 mins |
| JudgeLM | 7B | 1 | ✗ | ✗ | 24 mins |
| JudgeLM | 7B | 1 | ✓ | ✓ | 50 mins |
| JudgeLM | 7B | 1 | ✓ | ✗ | 3 mins |
| JudgeLM | 13B | 1 | ✓ | ✗ | 5 mins |
| JudgeLM | 33B | 2 | ✓ | ✗ | 15 mins |

Table 4: Performance analysis for the scaling JudgeLM on our *val* set.

| Judge Size | Data Scale | Agreement ↑ (w/ GPT-4) | Consistency ↑ (w/ swap.) | Bias ↓ toward 1st | Bias ↓ toward 2nd | Delta bias ↓ |
|---|---|---|---|---|---|---|
| 7B | 3.5k | 75.87 | 73.45 | 19.83 | 6.72 | 13.11 |
| 7B | 10k | 78.89 | 78.25 | 17.30 | 4.45 | 12.85 |
| 7B | 30k | 81.43 | 80.89 | 14.54 | 4.57 | 9.97 |
| 7B | 100k | 83.71 | 82.62 | 12.31 | 5.07 | 7.24 |
| 13B | 3.5k | 80.61 | 78.91 | 14.68 | 6.41 | 8.27 |
| 13B | 10k | 83.19 | 81.90 | 13.42 | 4.68 | 8.74 |
| 13B | 30k | 84.39 | 82.99 | 11.96 | 5.05 | 6.91 |
| 13B | 100k | 85.87 | 83.01 | 11.53 | 5.46 | 6.07 |
| 33B | 3.5k | 85.38 | 85.16 | 9.34 | 5.50 | 3.84 |
| 33B | 10k | 87.49 | 86.40 | 8.32 | 5.28 | 3.04 |
| 33B | 30k | 88.84 | 87.34 | 7.57 | 5.09 | 2.48 |
| 33B | 100k | 90.06 | 87.93 | 6.85 | 5.22 | 1.63 |

Table 5: Ablation study for the swap augmentation on our *val* set.

| Methods | Agreement ↑ (w/ GPT-4) | Consistency ↑ (w/ swap.) | Bias ↓ toward 1st | Bias ↓ toward 2nd | Delta Bias ↓ |
|---|---|---|---|---|---|
| baseline | 75.87 | 73.45 | 19.83 | 6.72 | 13.11 |
| + swap aug. | 76.51 | 78.89 | 15.34 | 5.77 | 9.57 |

tuned JudgeLM can outperform its teacher in this specific task. Besides, we also propose a human-annotated multimodal judging benchmark to evaluate our JudgeLM as shown in Table 13.

**Efficiency comparison.** To further compare the efficiency between our JudgeLM and PandaLM, we conduct experiments on our *val* set to display the time cost using the same machine with 8 NVIDIA-A100 (40G) GPUs. As shown in Table 3, we display the methods and model sizes in the first and second columns. The third column shows the needed GPUs for each judge model. The models with 7B or 13B parameters run on 1 A100 GPU with 40G memory while the 33B-parameter needs 2 GPUs. The fourth column shows whether the methods can judge answers in parallel. The fifth column indicates whether judge reasons are generated at runtime. The sixth column presents the total time cost. We use PandaLM-7B and JudgeLM-7B as efficiency baselines, which do not use parallel judging and generate detailed reasons for all questions. Thanks to the modeling of JudgeLM, i.e., "grading, judging, and reasoning", JudgeLM can skip the reasoning phase and only requires 24 minutes, which is 16.65 × faster than baselines. When we enable the engineering optimization of parallel judging, the JudgeLM can make full use of the 8 GPUs and cost only 50 minutes, which is 8 × faster than the baseline running on a single GPU. When we enable parallel judging and skip the reasoning phase for JudgeLM, the JudgeLM-7B only consumes 3 minutes to judge 5000 response pairs, which is 133.3 × faster than baselines. The largest judge model, JudgeLM-33B, can also complete the validation within 15 minutes. JudgeLM's high efficiency can significantly reduce the time spent on evaluating LLMs, allowing researchers and developers to boost the pace of advancements.

Table 6: Ablation study for the reference support and reference drop on our $val$ set.

| Methods | $ft$ w/ ref? | $val$ w/ ref? | Agreement ↑ (w/ GPT-4) | Consistency ↑ (w/ swap.) | Bias ↓ toward 1st | Bias ↓ toward 2nd | Delta Bias ↓ |
|---|---|---|---|---|---|---|---|
| *matching format.* | | | | | | | |
| baseline | ✗ | ✗ | 75.87 | 73.45 | 19.83 | 6.72 | 13.11 |
| baseline | ✓ | ✓ | 80.15 | 81.23 | 11.55 | 7.22 | 4.33 |
| *mismatched format.* | | | | | | | |
| baseline | ✗ | ✓ | 73.09 | 67.75 | 29.44 | 2.81 | 26.63 |
| baseline | ✓ | ✗ | 75.69 | 73.40 | 20.89 | 5.71 | 15.18 |
| *w/ ref. drop.* | | | | | | | |
| baseline | ref. drop | ✗ | 76.86 | 77.13 | 17.30 | 5.57 | 11.73 |
| baseline | ref. drop | ✓ | 80.35 | 81.24 | 11.48 | 7.28 | 4.20 |

Table 7: Performance of JudgeLM-7B with explanation-first (CoT) or score-first (Ours) on JudgeLM $val$ set.

| Methods | Agreement ↑ (w/ GPT-4) | Consistency ↑ (w/ swap.) | Bias ↓ toward 1st | Bias ↓ toward 2nd | Delta Bias ↓ |
|---|---|---|---|---|---|
| score-first (Our) | 75.87 | 73.45 | 19.83 | 6.72 | 13.11 |
| explanation-first (CoT) | 75.54 | 74.39 | 15.05 | 10.56 | 4.50 |

## 6.2 SCALING ANALYSIS OF JUDGELM

In this section, we analyze the scaling ability of the plain JudgeLM (without the proposed methods) on our $val$ set without reference as illustrated in Table 4. As we increase the model size and data scale, we can observe the metrics increase. It demonstrates that the proposed JudgeLM is scalable and can reach up to 90.06% agreement and 87.93% consistency with 33B-parameter and 100K fine-tuning data.

## 6.3 ABLATION STUDY

In this section, we present the ablation studies of the proposed methods. For all ablation studies, we use JudgeLM-7B as the base model and 3.5K data for fine-tuning. Based on this baseline, we analyze the improvements brought by swap augmentation, reference support, and reference drop.

**Improvements of Swap Augmentation.** As shown in Table 5, swap augmentation can improve the baseline model comprehensively. It improves consistency by 5.44%, which demonstrates that swap augmentation can reduce the influence of position bias and push the judge to pay more attention to the contents of answers.

**Improvements of Reference Support.** As shown in the rows with the matching format of Table 6, JudgeLM fine-tuned with reference support exhibits superior performance on every metric. It demonstrates that the introduction of reference answers induces the judge to rely on external knowledge and addresses the limitation of pre-trained knowledge.

**Improvements of Reference Drop.** As shown in Table 6, baselines can not reach satisfactory performance when facing mismatched formats. With the help of the reference drop, the JudgeLM can handle both the format with or without reference and achieve higher agreement and consistency. It demonstrates that reference drop can address the format bias and avoid the JudgeLM overfitting to a single format.

**Ablation of Judging Form** We further evaluate the performance of JudgeLM-7B with explanation-first (Chain of Thought, CoT (Wei et al., 2022)) or score-first (Ours) in Table 7. JudgeLM with CoT performs similar agreement with our score-first baseline but with higher consistency, which means that explanation-first form, i.e., CoT, can alleviate the position bias of fine-tuned judges, but not bring significant agreement improvement. As a result, we choose the score-first method for JudgeLM, which has slightly less consistency but more flexible usage.

Table 8: Comparison between GPT-4 teacher and JudgeLM-33B on JudgeLM $val$ set.

| Methods | Agreement ↑ (w/ GPT-4) | Consistency ↑ (w/ swap.) | Bias ↓ toward 1st | Bias ↓ toward 2nd | Delta Bias ↓ |
|---|---|---|---|---|---|
| GPT-4 | – | 85.82 | 6.10 | 8.10 | 2.00 |
| JudgeLM-33B | 89.03 | 91.36 | 5.55 | 3.09 | 2.46 |

### 6.4 ADDITIONAL EXPERIMENT

**Comparison with GPT-4 Teacher** As shown in Table 8, we further list the metrics except for the agreement with GPT-4 itself. JudgeLM-33B achieves higher consistency than GPT-4, which demonstrates that fine-tuning judges like JudgeLM-33B can achieve higher consistency through the proposed techniques.

The learning paradigm of JudgeLM is similar to knowledge distillation, as JudgeLM learns from the expert judgments provided by GPT-4. In the knowledge distillation domain (Gou et al., 2021), the student model (JudgeLM) mimics the teacher model (GPT-4) to achieve competitive or even superior performance. Additionally, our JudgeLM employs three key methods and utilizes large-scale training data specifically for the judging task to enhance its agreement and consistency. As mentioned in "Comparison on Other Human Evaluation Benchmarks" under Section 6.1, our experiments demonstrate that a fine-tuned specialist judge model can surpass its generalist teacher on some judging benchmarks, i.e., JudgeLM $val$ set and PandaLM $test$ set.

### 6.5 DETAILS OF DATASET

For the proposed dataset and benchmark, we also provide the explanation of usage scope, details of metric calculations, dataset quality, question category & distribution (among 19 categories), and comparison with UltraFeedback (Cui et al., 2023) in Sec. A.1. We hope the dataset can help researchers build more robust evaluation tools in the future.

### 6.6 GENERALIZATION ABILITY OF JUDGELM

Not only judging answer pairs, but our JudgeLM can also generalize to various judging tasks (including math problems and code generation), unseen judging benchmarks (human-annotated benchmark, multimodal judging benchmark, retrieval-format benchmark, multiple-format benchmark, toxic chat benchmark, reward model benchmark), and other various judging extensions (grading single answer, multi-turn chat). We leave the detailed analysis in Sec. A.3 of the appendix.

### 6.7 MORE DISCUSSION

Due to the limitation of pages, we leave more discussion in Sec. A.4 of the appendix.

## 7 CONCLUSION

In this paper, we first introduce a high-quality, large-scale dataset for LLM evaluation, that provides a robust foundation for future research. Next, the proposed JudgeLM as scalable judges for evaluating LLMs in open-ended tasks efficiently, achieving state-of-the-art judge performance on two benchmarks. Then, we analyze two key biases and introduce a new format bias in fine-tuning LLMs as judges, and address them with the proposed techniques. We hope our work can motivate more studies to explore the judge models for LLMs in open-ended tasks and build more powerful LLMs with guidance from judge models.

**Limitations.** Although the proposed JudgeLM achieves encouraging performance and efficiency, the cost of the judge dataset limits further scaling up in the judge dataset. Currently, we spend about 4000 dollars to provide 100K high-quality GPT-4-generated judge data to the public. We expect to further improve the performance of judge models with the help of synthetic judge data.

ACKNOWLEDGMENTS AND DISCLOSURE OF FUNDING

This work was partially supported by the National Natural Science Foundation of China (NSFC) under Grant No. 62276108.

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

## A  APPENDIX / SUPPLEMENTAL MATERIAL

### A.1  MORE ABOUT DATASET

**Dataset Usage Scope** We emphasize that the JudgeLM dataset is intended only for academic research and any commercial use is prohibited. Because the OpenAI's terms prohibit developing models that compete with OpenAI, the instruction-tuning datasets generated by the OpenAI's API, i.e., Alpaca, PandaLM, etc., all follow this rule.

**Details of Metric Calculations** For objective metrics, we use the judgments annotated by humans or GPT-4 as ground truth labels, and the judgments generated by judge models as predicted labels. We use $TP$, $FP$, $TN$, and $FN$ to represent the true positive, false positive, true negative, and false negative, respectively. The calculation of agreement, precision, recall, and F1-score are as follows:

$$\text{Agreement} = (TP + TN)/(TP + FP + TN + FN), \tag{1}$$

$$\text{Precision} = TP/(TP + FP), \tag{2}$$

$$\text{Recall} = TP/(TP + FN), \tag{3}$$

$$\text{F1-score} = (2 * TP)/(2 * TP + FP + FN). \tag{4}$$

For reliability metrics, we compare the results before and after swapping the order of the two answers (A and B), i.e., from "the first answer is A, the second answer is B" to "the first answer is B, the

second answer is A". When the judging results change, we mark it as a biased sample. Samples with a bias toward the first position consist of three situations, "from Answer A wins to tie", "from Answer A wins to Answer B wins", and "from tie to Answer B wins". Similarly, samples with a bias toward the second position also consist of three situations, "from Answer B wins to tie", "from Answer B wins to Answer A wins", and "from tie to Answer A wins". The calculation of metrics of "bias toward 1st", "bias toward 2nd", "delta bias" are defined as follows:

$$\text{bias toward 1st} = \frac{\text{Number of samples with a bias toward the first position}}{\text{Numbers of total samples}}, \quad (5)$$

$$\text{bias toward 2nd} = \frac{\text{Number of samples with a bias toward the second position}}{\text{Numbers of total samples}}, \quad (6)$$

$$\text{delta bias} = |\text{bias toward 1st} - \text{bias toward 2nd})|. \quad (7)$$

**Dataset Quality** To ensure the high quality of the proposed dataset, we filter the low-quality data samples in each step. For data samples (including seed tasks and references) in four public datasets, i.e., Alpaca-GPT4 (Peng et al., 2023), Dolly-15K (Conover et al., 2023), GPT4All-LAION (Anand et al., 2023), and ShareGPT, we first remove data samples containing obviously incorrect, irrelevant, or harmful reference answers through automated filtering scripts. Next, we randomly sample 105K samples from the filtered set and extract answers from 11 LLMs. Then, we input the tasks, randomly sampled answer pairs, and optionally reference answers to the GPT-4 teacher for judgment. Finally, the authors of this work are involved in a multi-step validation process to ensure the quality, accuracy, and reliability of the judge samples. This process includes an initial annotation step where GPT-4 provides preliminary judgments, followed by independent human re-annotation where authors provide simple judgments ("Answer 1 wins," "Answer 2 wins", or "Tie") without exposure to GPT-4's annotations. The final step involves cross-validation and refinement, where human judgments are compared with GPT-4 annotations to thoroughly verify the judge results, scores, and reasoning quality. Please note that incorporating high-quality answers from closed-sourced models e.g., Qwen (Bai et al., 2023) and Claude, could enhance the diversity of the dataset. We leave it as a future work.

**Question Category & Distribution of Validation Set** We count the distribution of questions in the JudgeLM $val$ set as shown in Table 9. Please note that the question categories included in the JudgeLM $val$ set and $train$ set are the same, but none of the data samples are identical.

**Comparison with UltraFeedback** Furthermore, we compare the JudgeLM dataset with UltraFeedback (Cui et al., 2023), which is an excellent dataset serving as a solid foundation for feedback-learning research. JudgeLM has nearly half more seeds than UltraFeedback, and an additional validation set containing 5K seeds. JudgeLM and UltraFeedback both provide scalar and text feedback, GPT-4 annotation, and fine-grained consideration. However, the JudgeLM dataset is further checked and re-annotated by humans, which provides double-checking on the quality of feedback. Moreover, JudgeLM supports judging with references, which can make up for the lack of pre-trained knowledge or insert specific judge preferences. Finally, JudgeLM clearly splits the seeds into 19 categories providing intuitive ability estimation for judges.

| Dataset | $train$ Seeds | $val$ Seeds | Feedback Format | Annotator | fine grained? | with Ref.? | Seeds Categories |
|---------|------------|----------|-----------------|-----------|---------------|------------|------------------|
| UltraFeedback | 64K | 0 | Scalar & Text | GPT-4 | Y | N | - |
| JudgeLM | 100K | 5K | Scalar & Text | GPT-4 & Human | Y | Y | 19 |

**Comparison with PandaLM Test Set** Furthermore, we compare the JudgeLM dataset with the PandaLM test set. An analysis of task distributions in Table 10 shows significant differences between the PandaLM test set and the JudgeLM benchmark. For example, business, fact-QA, summarizing, linguistics, emotion, entity-processing, explain, retrieval, document, and chat are well-represented in PandaLM but absent in JudgeLM, while writing and roleplay show a significant delta percentage (over 4%). This confirms that the PandaLM test set includes 49% unseen task samples that are out of distribution for JudgeLM.

Table 9: Distribution of question categories in JudgeLM *val* set

|  | count | percentage |  | count | percentage |
|---|---|---|---|---|---|
| culture | 233 | 4.66% | planning | 309 | 6.18% |
| recommendation | 482 | 9.64% | roleplay | 77 | 1.54% |
| finance | 142 | 2.84% | coding | 201 | 4.02% |
| science | 393 | 7.86% | health | 278 | 5.56% |
| technique | 42 | 0.84% | writing | 625 | 12.50% |
| common-sense | 373 | 7.46% | hardware | 130 | 2.60% |
| art | 335 | 6.70% | history | 243 | 4.86% |
| math | 250 | 5.00% | geography | 199 | 3.98% |
| private-matter | 421 | 8.42% | others | 63 | 1.26% |
| law | 204 | 4.08% | total | 5000 | 100.00% |

Table 10: Distribution of question categories in PandaLM test set. The $\Delta$ represents the percentage difference compared to the JudgeLM benchmark. We **bolded** categories that appear in the PandaLM test set but don't exist in the JudgeLM benchmark.

|  | count | percentage | $\Delta$ |  | count | percentage | $\Delta$ |
|---|---|---|---|---|---|---|---|
| **business** | 87 | 8.71% | 8.71% | writing | 81 | 8.11% | -4.39% |
| **fact-QA** | 70 | 7.01% | 7.01% | planning | 57 | 5.71% | -0.47% |
| **summarizing** | 64 | 6.41% | 6.41% | roleplay | 57 | 5.71% | 4.17% |
| **linguistics** | 45 | 4.50% | 4.50% | coding | 54 | 5.41% | 1.39% |
| **emotion** | 45 | 4.50% | 4.50% | art | 44 | 4.40% | -2.30% |
| **entity-processing** | 42 | 4.20% | 4.20% | finance | 40 | 4.00% | 1.16% |
| **explain** | 41 | 4.10% | 4.10% | culture | 38 | 3.80% | -0.86% |
| **retrieval** | 40 | 4.00% | 4.00% | math | 37 | 3.70% | -1.30% |
| **document** | 30 | 3.00% | 3.00% | geography | 12 | 1.20% | -2.78% |
| **chat** | 26 | 2.60% | 2.60% | others | 5 | 0.50% | -0.76% |
| recommendation | 84 | 8.41% | -1.23% | total | 999 | 100.00% |  |

## A.2 FINE-TUNING SETTING

We list the hyper-parameters we used, as shown in Table 11.

## A.3 GENERALIZATION ABILITY OF JUDGELM

To validate the generalization ability of JudgeLM, we test JudgeLM on various judging tasks (including math problems and code generation), unseen judging benchmarks (human-annotated benchmark, multimodal judging benchmark, retrieval-format benchmark, multiple-format benchmark), and other various judging extensions (grading single answer, multi-turn chat).

**Generalize to Various Judging Tasks.** To further validate the judging performance on questions with specific categories, we present the judging results of JudgeLM-33B on these questions, i.e., coding, common-sense, math, roleplay, and writing. Table 12 shows that JudgeLM can handle the judging tasks of various categories. We also find that the judging performance of math questions is slightly lower than coding and common-sense questions. However we think it is a common problem of large language models (Imani et al., 2023), and future advancement on base models (Yu et al., 2023a; Luo et al., 2023) would alleviate this problem.

**Generalize to Multimodal Judging Benchmark.** Traditional multimodal evaluation needs prediction to match the ground truth exactly. For some open-ended questions, a human-like evaluator is

Table 11: JudgeLM fine-tuning setting.

| config | JudgeLM / -7B / -13B / -33B |
|---|---|
| base model | Vicuna / -7B / -13B / -33B |
| model max length | 2048 |
| fine-tuning data source | JudgeLM-100K |
| learning rate | 2e-5 |
| learning rate schedule | cosine decay |
| optimizer | AdamW (Kingma & Ba, 2014; Loshchilov & Hutter, 2019) |
| optimizer hyper-parameters | $\beta_1, \beta_2, \epsilon = 0.9, 0.999, 1e\text{-}8$ |
| weight decay | 0.0 |
| GPU nums | 8 / 8 / 16 |
| batch size | 128 |
| training epochs | 3 |
| warmup ratio | 0.003 |
| numerical precision | bf16, tf32 |
| ZeRO optimizer (Ramesh et al., 2021) | stage 3 |
| gradient checkpointing | True |
| GPT-3.5 and GPT-4 version | 2023-03-15-preview |

Table 12: Performance of JudgeLM-33B with specific categories on JudgeLM $val$ set.

| | Agreement ↑ (w/ GPT-4) | Consistency ↑ (w/ swap.) | Bias ↓ toward 1st | Bias ↓ toward 2nd | Delta Bias ↓ |
|---|---|---|---|---|---|
| ***coding*** | | | | | |
| val w/o ref | 88.08 | 88.60 | 6.22 | 5.18 | 1.04 |
| val w/ ref | 88.83 | 91.37 | 3.55 | 5.08 | 1.53 |
| ***common-sense*** | | | | | |
| val w/o ref | 88.41 | 90.43 | 7.25 | 2.32 | 4.93 |
| val w/ ref | 90.37 | 92.35 | 4.53 | 3.12 | 1.41 |
| ***math*** | | | | | |
| val w/o ref | 86.45 | 84.08 | 7.76 | 8.16 | 0.40 |
| val w/ ref | 86.81 | 87.85 | 4.45 | 7.69 | 3.24 |
| ***croleplay*** | | | | | |
| val w/o ref | 88.00 | 88.00 | 6.67 | 5.33 | 1.34 |
| val w/ ref | 88.72 | 89.12 | 5.01 | 5.87 | 0.86 |
| ***writting*** | | | | | |
| val w/o ref | 87.33 | 92.36 | 3.25 | 4.39 | 1.14 |
| val w/ ref | 89.11 | 92.83 | 3.67 | 3.50 | 0.17 |

needed to determine whether the prediction is close to the ground truth range. Modern multimodal, such as MM-Vet (Yu et al., 2023b) and Prometheus-Vision (Lee et al., 2024), which use GPT-4V, GPT-4 or GPT-3.5 as judges. The API-based judge takes the question text, ground-truth text, the model's prediction, and optional input image as input, and makes judgments based on them. Our JudgeLM also provides good practice for such a multimodal evaluation by a slightly modified template as shown in Fig. 7. Thanks to its capacity to judge open-ended answers, our JudgeLM can also perform well in judging multimodal models, as shown in Fig. 21.

We further conduct experiments to evaluate JudgeLM' ability to judge multimodal models when compared with close-sourced LLM, i.e., GPT-3.5 and GPT-4. We first use GPT-4, GPT-3.5, and JudgeLM to judge the LLaVA's output (Liu et al., 2023), respectively. Then, we collect judgments from human annotators, whose judgments include three situations: completely correct, semi-correct, and completely wrong. Last, we compute the metrics between the LLM judges' judgments and human judgments, as shown in Table 13. It can be observed that JudgeLM outperforms GPT-4 (0-shot) and GPT-3.5 (7-shot). Besides, JudgeLM achieves 2.5% higher precision than GPT-4 (7-shot). The encouraging results demonstrate the generalization ability of JudgeLM in dealing with multimodal judging. Furthermore, JudgeLM can use large multimodal models, e.g., LLaVA (Liu et al., 2023), as the backbone for better processing the multimodal judging. We leave it as a future work.

Table 13: JudgeLM zero-shot evaluation results on human-annotated MM-Vet benchmark.

| Methods | Agreement ↑ (w/ Human) | Precision ↑ (w/ Human) | Recall ↑ (w/ Human) | F1 ↑ (w/ Human) |
|---|---|---|---|---|
| GPT-4 (7-shot) | 95.58 | 88.63 | 87.79 | 88.04 |
| GPT-4 (0-shot) | 86.70 | 79.75 | 86.41 | 81.81 |
| GPT-3.5 (7-shot) | 83.03 | 76.14 | 74.84 | 73.62 |
| JudgeLM-33B (0-shot) | 91.74 | 91.08 | 85.58 | 87.26 |

**Generalize to Out-of-distribution ToxicChat Benchmark.** To further evaluate the generalization ability of the proposed JudgeLM, we selected an out-of-distribution benchmark, i.e., Toxic-Chat (Lin et al., 2023), for evaluation. Following the guidelines provided in the ToxicChat dataset, we conducted experiments on the latest test set (0124) of ToxicChat, comparing OpenAI Moderation, the GPT-4 teacher, and our proposed JudgeLM. OpenAI Moderation is an API trained on publicly available toxicity datasets, primarily sourced from social media. For both the GPT-4 teacher and JudgeLM, we used the same templates and thresholds. As shown in Table 14, our proposed JudgeLM achieves superior precision and comparable accuracy to the specialist model, i.e., OpenAI Moderation. These results demonstrate that JudgeLM can further generalize to out-of-distribution datasets such as ToxicChat.

Table 14: JudgeLM zero-shot evaluation results on toxic-chat test set.

| Methods | Accuracy↑ | Precision↑ | Recall↑ | F1↑ |
|---|---|---|---|---|
| *Specialist API-based Method* | | | | |
| OpenAI Moderation | 89.70 | 54.76 | 69.89 | 61.41 |
| *Generalist API-based Method* | | | | |
| GPT-4 | 88.08 | 52.17 | 73.20 | 60.92 |
| *Open-sourced Method* | | | | |
| JudgeLM-33B | 89.66 | 58.79 | 61.88 | 60.30 |

**Generalize to Retrieval-format Benchmark.** In real-world applications, we do not always have well-organized reference answers for judging. To evaluate the capability of JudgeLM in dealing with this situation, we inject the original reference answers into a randomly selected paragraph. As shown in Table 15, we select paragraphs with different words, and evaluate JudgeLM-33B with the injected paragraphs as references in a zero-shot setting. The results show that JudgeLM can retrieve the correct answers from the paragraphs and make judgments based on them. When the words of paragraphs increase to 400, JudgeLM faces a maximum drop of 3.73% agreement and 3.37%

consistency. The results demonstrate that JudgeLM is promising for utilizing JudgeLM to deal with unstructured references or judge in the retrieved form.

Table 15: Performance of JudgeLM-33B with injected paragraphs as references on JudgeLM *val* set.

| Reference Paragraph | Agreement ↑ (w/ GPT-4) | Consistency ↑ (w/ swap.) | Bias ↓ toward 1st | Bias ↓ toward 2nd | Delta Bias ↓ |
|---|---|---|---|---|---|
| No | 89.32 | 92.37 | 3.62 | 4.01 | 0.39 |
| 50 words | 87.78 | 92.35 | 3.00 | 4.65 | 1.65 |
| 100 words | 87.69 | 91.84 | 2.62 | 5.54 | 2.92 |
| 200 words | 86.77 | 90.48 | 2.70 | 6.82 | 4.12 |
| 300 words | 86.25 | 89.26 | 3.13 | 7.61 | 4.48 |
| 400 words | 85.59 | 89.00 | 2.98 | 8.02 | 5.04 |

**Generalize to Multiple-format benchmark.** To get the optimal ranking for N answers from different LLMs, other judge models need to call the model $O(n^2)$ times to get the full matrix, which is a much less efficient solution. We attempt to resolve this limitation by extending our JudgeLM to process multiple answers at the same time. We first need to modify the template as shown in Fig. 5. As shown in Fig. 18, JudgeLM can judge and rank the multiple answers within the context limit of LLM.

We further conduct experiments to evaluate the consistency in the judging form of answer pairs and multiple answers. We first generate answers on JudgeLM *val* set through 3 LLMs, i.e., Vicuna-13B, LLaMA-7B, and alpaca-7B, for evaluation. Then we use pairwise judging and multiple judging to grade answers and rank them, respectively. Last, we compute the consistency between the two ranking results. Please note that 'Error Rate@2' indicates the position orderings of two answers are different between the result of paired judgment and the result of multiple judgment, and 'Error Rate@3' means the position orderings of three answers are different. Table 16 shows that consistency between judging pairwise and judging multiple can reach 93.48%, and only 0.14% results are totally wrong. The results are impressive but the 6.38% of 'Error Rate@2' also shows room for improvement as well, which could be addressed by the further improvement of JudgeLM's self-consistency.

Table 16: Performance of JudgeLM-33B in judging multiple answers on JudgeLM *val* set. We calculate the consistency between the pairwise judging results and multiple judging ones.

| | Consistency ↑ (w/ pairwise) | Error Rate@2 ↓ (w/ pairwise) | Error Rate@3 ↓ (w/ pairwise) |
|---|---|---|---|
| JudgeLM-33B multipile | 93.48 | 6.38 | 0.14 |

**Generalize to Single Answer Grading.** The Concurrent judge method (Wang et al., 2023) usually judges a pair of answers to decide which one is better or tie but they lack the ability to evaluate a single answer. Thanks to our judging mode of scoring first and then calculating the judging results, our JudgeLM provides an alternative practice to grade a single answer by slightly modifying the template as shown in Fig. 5. Putting the reference answer in the first position and giving it a full grade as a prior, JudgeLM can give quantitative fine-grained evaluations as shown in Fig. 17.

The capability of grading a single answer is an important extension, which only relies the text-form prediction and ground truth to make judgments. For example, the following extension "Judging multimodal models" is also based on this capability.

**Generalize to Reward Model.** We further compare the proposed JudgeLM with closed-source methods, powerful reward models, Llama-2-based reward methods (Touvron et al., 2023b; Ivison et al., 2023), and Qwen-1.5-based reward methods (Bai et al., 2023) on the requested reward model benchmark as shown in Table 17. Following the evaluation manner of GPT-3.5-turbo-0125 and Prometheus series (Kim et al., 2023) in RewardBench (Lambert et al., 2024), we evaluate the models among 4 subsets, i.e., Chat, (Chat) Hard, Safety, and Reasoning, and present averaged score in the Score column. The proposed JudgeLM-7B outperforms the GPT-3.5-turbo-0125, the Prometheus series reward models (Kim et al., 2023), UltraRM-13B (Cui et al., 2023), TIGERScore-

Table 17: Comparison with advanced reward models on RewardBench.

| Model | Score | Chat | Hard | Safety | Reason |
|---|---|---|---|---|---|
| ***Closed-source Models*** | | | | | |
| GPT-3.5-turbo-0125 | 64.5 | 92.2 | 44.5 | 62.3 | 59.1 |
| ***Powerful Reward Models.*** | | | | | |
| Prometheus-8×7B-v2.0 | 75.3 | 93 | 47.1 | 83.5 | 77.4 |
| Prometheus-7B-v2.0 | 72.5 | 85.5 | 49.1 | 78.7 | 76.5 |
| UltraRM-13B | 67.6 | 96.4 | 55.5 | 56.0 | 62.4 |
| TIGERScore-13B | 35.6 | 35.2 | 32.9 | 41.5 | 32.7 |
| ***Llama-2-based Models.*** | | | | | |
| Tulu-2-dpo-70B | 79.0 | 97.5 | 60.5 | 83.9 | 74.1 |
| Tulu-2-dpo-13B | 76.4 | 95.8 | 58.3 | 78.2 | 73.2 |
| Tulu-2-dpo-7B | 74.7 | 97.5 | 56.1 | 73.3 | 71.8 |
| ***Qwen1.5-based Models.*** | | | | | |
| Qwen1.5-72B-Chat | 71.5 | 62.3 | 66.0 | 72.0 | 85.5 |
| Qwen1.5-14B-Chat | 73.4 | 57.3 | 70.2 | 76.3 | 89.6 |
| Qwen1.5-7B-Chat | 72.0 | 53.6 | 69.1 | 74.8 | 90.4 |
| ***Ours.*** | | | | | |
| JudgeLM-7B | 78.5 | 92.2 | 56.1 | 83.2 | 82.3 |

13B (Jiang et al., 2023), Tulu-2-dpo-13B (Ivison et al., 2023), Tulu-2-dpo-7B (Ivison et al., 2023), and Qwen1.5-based reward models (Bai et al., 2023). Besides, JudgeLM-7B even achieves similar performance to Tulu-2-dpo-70B (Ivison et al., 2023). Noting the RewardBench paper mentions that involving Llama-3 as the base model can significantly improve the metrics on Hard and Reasoning, we leave it as future work.

**Multi-turn Chat about Judgments.** It is worth noting that fine-tuning with judge samples does not compromise the multi-turn chat ability extended from base models. As illustrated in Fig. 19 and Fig. 20, our JudgeLM retains the capability to engage in meaningful dialogues with users, providing them with a richer context, detailed information, additional examples, and specific details.

## A.4 MORE DISCUSSION

Table 18: Comparison of different base models for JudgeLM-7B on JudgeLM *val* set.

| Base Models (for JudgeLM-7B) | Agreement ↑ (w/ GPT-4) | Precision ↑ (w/ GPT-4) | Recall ↑ (w/ GPT-4) | F1 ↑ (w/ GPT-4) | Consistency ↑ (w/ swap.) |
|---|---|---|---|---|---|
| ***Judge w/o reference.*** | | | | | |
| Vicuna | 81.11 | 69.67 | 78.39 | 72.21 | 83.57 |
| LLaMA2-chat | 83.87 | 73.43 | 80.06 | 75.91 | 85.17 |
| ***Judge w/ reference.*** | | | | | |
| Vicuna | 84.08 | 75.92 | 82.55 | 78.28 | 84.46 |
| LLaMA2-chat | 86.60 | 79.47 | 83.11 | 81.02 | 87.74 |

**Format Bias** As shown in Table 16, it can be seen that the judging of multiple answers does not receive a significant performance drop. We hold the viewpoint that judging multiple answers is an easy extension for JudgeLM, which does not change the basis for judging. As mentioned in '4 Inherent Biases - Format Bias', format bias means the model judging basis changes from pre-trained knowledge to reference, or vice versa. So, judging in mismatched situations faces format bias but judging multiple answers does not.

**Other Human-annotated benchmarks** For a fair comparison, we also evaluate JudgeLM on the PandaLM *test* set in a zero-shot setting. The PandaLM *train* and *test* sets are annotated by GPT3.5

and humans, respectively. As shown in Table 2, the zero-shot results of JudgeLM also outperform other judging methods, i.e., PandaLM, GPT-3.5, and GPT-4. Furthermore, JudgeLM also achieves a superior 0-shot judging performance on the multimodal benchmark with human annotation, i.e., MM-Vet, as shown in Table 13.

**Reasoning Ability of LLM Judges** Nowadays, NLP researchers are still struggling with proposing LLMs with superior reasoning abilities. JudgeLM also needs the proposed reference sup method to enhance the judging ability for out-of-domain or counterfactual tasks, as shown in Fig. 11 and Fig. 16. Notably, the proposed JudgeLM can benefit from stronger foundation LLMs, e.g., the LLaMA2-7B-Chat-based (Touvron et al., 2023b) JudgeLM outperforms the original JudgeLM-7B on all metrics, as shown in Table 18. The research of judge models is critical for the development of LLMs and can benefit from advanced LLMs, establishing a positive cycle.

**Critiques for Judgements** Beyond using LLMs to compare answer pairs and generate reasons, the critique and correction of these reasons have become increasingly important topics. This approach allows LLMs to reassess their generated reasons in multiple rounds, thereby enhancing judging accuracy. Works like UltraCM, Auto-J, and Shepherd have reliably evaluated the quality of textual reasons. Recently, CriticBench has also provided a reliable benchmark to evaluate the reasons and evaluation abilities of LLMs.

For JudgeLM, its ability to generalize to multi-turn chat enables us to construct multi-turn judgement critique data by combining data samples without references and those with references. Through two rounds of judge Q&A, i.e., without reference and with reference, JudgeLM can acquire the capability to critique its own judgments. We leave these experiments for future work.

**Reference Drop as an Independent Method** We think the reference drop is an independent and significant method. At first, we argue that judging with or without references are two sub-benchmarks, which require judges to make judgments with internal knowledge or by comparing LLM-generated answers with a reference answer, respectively. The reference drop is not only a simple but effective hyper-parameter, but also an important method that bridges the two sub-benchmarks, which enables the JudgeLM to make judgments in different situations.

Table 19: Comparison between PandaLM and JudgeLM components in terms of datasets and methods.

| Base Model | Data | Method | val w/ ref? | Agreement ↑ | Consistency ↑ |
|---|---|---|---|---|---|
| ***PandaLM baseline.*** | | | | | |
| LLaMA | + 300K PandaLM data | | N | 68.61 | 74.78 |
| ***change to JudgeLM data.*** | | | | | |
| LLaMA | +3.5k JudgeLM data | | N | 71.30 | 69.59 |
| ***w/ the proposed methods.*** | | | | | |
| LLaMA | +3.5k JudgeLM data | +swap aug | N | 72.41 | 73.50 |
| LLaMA | +3.5k JudgeLM data | +ref sup | Y | 75.15 | 74.08 |
| LLaMA | +3.5k JudgeLM data | +ref drop | Y | 75.93 | 74.77 |
| ***w/ all proposed methods.*** | | | | | |
| LLaMA | +3.5k JudgeLM data | + all | N | 75.01 | 76.28 |
| LLaMA | +3.5k JudgeLM data | + all | Y | 78.10 | 79.50 |

**Differences between PandaLM and JudgeLM** To fairly assess the impact of our proposed dataset and methods, we conducted experiments using the PandaLM baseline with LLaMA-7B as the base model. As shown in Table 19, our dataset and methods both provide significant improvements.

Compared with PandaLM, our method has these different novelties:

- We introduce **a high-quality, large-scale dataset for judge models**, enriched with diverse seed tasks, LLMs-generated answers, and detailed judgments from GPT-4, laying the foundation for future LLMs evaluating research.

- We analyze the **biases inherent to LLM judge fine-tuning** and introduce **a series of methods** to address them. Our methods significantly improve the consistency of the model in different cases, making the JudgeLM more reliable and flexible.

- The proposed **judging pattern, i.e., grading, judging, and reasoning**, makes judging efficient, which only needs little time to grade and judge, and generates time-consuming reasons optionally.

Furthermore, other contributions which differ from PandaLM are as follows:

- We analyze the **finer scaling ability** of the language model judge, and the scales of training data, i.e., JudgeLM-7B, JudgeLM-13B, JudgeLM-33B, 3.5k-data, 10k-data, 30k-data, and 100k-data, for evaluating LLMs in open-ended scenarios. The JudgeLM-33B-100K even **exceeds the humans' judging agreement**.

- The proposed JudgeLM presents **extended capabilities** as shown in Fig. 1b, including grading single answers, judging multiple answers, judging multimodal models, multi-turn chat, and judging with injected paragraphs.

Specifically, the proposed JudgeLM presents generalization ability to seven different judging areas:

- Generalize to **19 various judging tasks**, including coding, common-sense, math, roleplay, writing, etc.

- Generalize to **human-annotated benchmarks**, such as the PandaLM test set and MM-Vet benchmark.

- Generalize to **multimodal judging benchmark**, such as MM-Vet benchmark.

- Generalize to **retrieval-format benchmark** which injects original reference answers into paragraphs with different words.

- Generalize to **multiple-format benchmark** which judges multiple answers at the same time.

- Generalize to **single-answer grading**.

- Generalize to **multi-turn chat about judgments**.

**GPT-4 Distilled Data** Evaluating Large Language Models (LLMs) in open-ended scenarios is challenging because existing benchmarks and metrics can not measure them comprehensively. Because the GPT-4-based judge can judge LLMs like a human expert, much concurrent work delving into curating training data (Li et al., 2023a; Kim et al., 2023; Cui et al., 2023; Xu et al., 2023; Jiang et al., 2023; Wang et al., 2023) and benchmarks (Lambert et al., 2024; Zheng et al., 2023) with the help of the closed-source LLMs, such as GPT-4 and GPT-3.5.

However, the GPT-4 teacher also faces inherent biases. Firstly, GPT-4's training data may contain cultural, societal, and linguistic biases. These biases may influence the GPT-4's judgment and lead to skewed evaluations. To mitigate these possible biases, the authors of this work are involved in double-checking to ensure the judgments from the GPT-4 teacher are accurate, objective, and unbiased. Secondly, the GPT-4's judgments also contain position bias and knowledge bias (Zheng et al., 2023). To address these problems, we introduce swap augmentation and reference support in fine-tuning LLMs as judges, which significantly improves the consistency and accuracy of fine-tuned judges.

**Does JudgeLM favor GPT-4?** Thank you for discussing this interesting phenomenon. We carefully conducted experiments between GPT-4 and GPT-3.5 to analyze the potential favoring of GPT-4 answers by JudgeLM. As shown in Table 20, GPT-4 wins 59.0% of pairwise comparisons, with only 16.2% of wins for GPT-3.5. We further conducted a sampling analysis to identify the following contributing factors:

- Alignment with Prompt Requirements: As shown in the judging templates (Fig. 3, Fig. 4, Fig. 5, Fig. 6, and Fig. 7), JudgeLM evaluates responses based on structured criteria, such as relevance and level of detail. Our analysis shows that GPT-4 answers tend to align better with these requirements compared to GPT-3.5 answers.

- Inherent Quality Difference: A sampling analysis of 500 cases (approximately 16.9% of the GPT-4 win cases) showed that in 482 instances (96.4%), GPT-4 answers were more accurate, detailed, and contextually relevant than GPT-3.5 answers. This analysis underscores that GPT-4 win cases are overwhelmingly attributed to the higher answer quality, rather than the preference from JudgeLM.

Moreover, the sample analysis shows that when the quality of the answers to GPT-4 and GPT-3.5 are similar, JudgeLM tends to give similar scores, resulting in a Tie. These results indicate that GPT-4 answers win more due to their better relevance, high level of detail, and high quality.

Table 20: Quantitative comparison results between GPT-4 and GPT-3.5.

|  | GPT4 win | Tie | GPT3.5 win |
|---|---|---|---|
| GPT4 v.s. GPT-3.5 | 59.00% | 24.80% | 16.20% |

**Reliability of GPT-4-annotated Data.** We conduct more experiments to further evaluate the reliability of GPT-4's annotation. In this experiment setting, we set annotations provided by Human 1 in our benchmark as ground truth. Then we calculate the agreement of JudgeLM-33B, GPT-4, and Human 2 as shown in Table 21. These results demonstrate that GPT-4 achieves higher agreement with the ground truth compared to Human 2. llm-as-a-judge (Zheng et al., 2023) shows similar results, which means GPT-4 teacher's judgments can align with human evaluators closely and can serve as a human-like teacher judge to provide high-quality and reliable judgments.

Table 21: Agreement evaluation of JudgeLM-33B, GPT-4, and Human, with another Human's judgments as ground truth.

|  | JudgeLM-33B | GPT-4 | Human 2 |
|---|---|---|---|
| Ground Truth (Human 1) | 90.72 | 84.48 | 79.82 |

**Future Work about Training Queries Distribution.** The distribution of training queries is a key factor in shaping the evaluation capabilities of Judge LLMs. While our current dataset was designed to ensure task diversity, MixEval (Ni et al., 2024b) and MixEval-X (Ni et al., 2024a) give us more insight, i.e., aligning the query distribution more closely with real-world user queries could further enhance the model's fairness and relevance. We leave it as a promising future work.

### A.5    PROMPT TEMPLATES

We list all the prompt templates we used.

### A.6    CASE STUDIES

We list several case studies.

You are a helpful and precise assistant for checking the quality of the answer.
[Question]
{question}

[The Start of Assistant 1's Answer]
{answer_1}
[The End of Assistant 1's Answer]

[The Start of Assistant 2's Answer]
{answer_2}
[The End of Assistant 2's Answer]

[System]
We would like to request your feedback on the performance of two AI assistants in response to the user question displayed above.
Please rate the helpfulness, relevance, accuracy, level of details of their responses. Each assistant receives an overall score on a scale of 1 to 10, where a higher score indicates better overall performance.
Please first output a single line containing only two values indicating the scores for Assistant 1 and 2, respectively. The two scores are separated by a space. In the subsequent line, please provide a comprehensive explanation of your evaluation, avoiding any potential bias and ensuring that the order in which the responses were presented does not affect your judgment.

Figure 3: The template for judging answers without the reference.

You are a helpful and precise assistant for checking the quality of the answer.
[Question]
{question}

[Reference Answer]
{reference}

[The Start of Assistant 1's Answer]
{answer_1}
[The End of Assistant 1's Answer]

[The Start of Assistant 2's Answer]
{answer_2}
[The End of Assistant 2's Answer]

[System]
We would like to request your feedback on the performance of two AI assistants in response to the user question displayed above.
Based on the reference answer, please rate the helpfulness, relevance, accuracy, level of details of their responses. Each assistant receives an overall score on a scale of 1 to 10, where a higher score indicates better overall performance.
Please first output a single line containing only two values indicating the scores for Assistant 1 and 2, respectively. The two scores are separated by a space. In the subsequent line, please provide a comprehensive explanation of your evaluation, avoiding any potential bias and ensuring that the order in which the responses were presented does not affect your judgment.

Figure 4: The template for judging answers with the reference.

You are a helpful and precise assistant for checking the quality of the answer.
[Question]
{question}

[The Start of Assistant 1's Answer]
{reference}
[The End of Assistant 1's Answer]

[The Start of Assistant 2's Answer]
{single answer}
[The End of Assistant 2's Answer]

[System]
We would like to request your feedback on the performance of two AI assistants in response to the user question displayed above.
Please rate the helpfulness, relevance, accuracy, level of details of their responses. Each assistant receives an overall score on a scale of 1 to 10, where a higher score indicates better overall performance.
Please first output a single line containing only two values indicating the scores for Assistant 1 and 2, respectively. The two scores are separated by a space. In the subsequent line, please provide a comprehensive explanation of your evaluation, avoiding any potential bias and ensuring that the order in which the responses were presented does not affect your judgment.

[Response]
10

Figure 5: The template for grading a single answer. We set the reference answer in the position of Answer 1. Then, we set the score of the reference answer to 10. Last, the JudgeLM outputs the score of the single answer with such a prior.

You are a helpful and precise assistant for checking the quality of the answer.
[Question]
{question}

[The Start of Assistant 1's Answer]
{answer_1}
[The End of Assistant 1's Answer]

[The Start of Assistant 2's Answer]
{answer_2}
[The End of Assistant 2's Answer]

[The Start of Assistant 3's Answer]
{answer_3}
[The End of Assistant 3's Answer]

[System]
We would like to request your feedback on the performance of three AI assistants in response to the user question displayed above.
Please rate the helpfulness, relevance, accuracy, level of details of their responses. Each assistant receives an overall score on a scale of 1 to 10, where a higher score indicates better overall performance.
Please first output a single line containing only three values indicating the scores for Assistant 1, 2 and 3, respectively. The three scores are separated by a space. In the subsequent line, please provide a comprehensive explanation of your evaluation, avoiding any potential bias and ensuring that the order in which the responses were presented does not affect your judgment.

Figure 6: The template for judging multiple answers.

Please evaluate the following visual-based question-answer pair:

[Question]
{question}

[Reference Answer]
{reference}

[The Start of Assistant 1's Answer]
{answer_1}
[The End of Assistant 1's Answer]

[The Start of Assistant 2's Answer]
{answer_2}
[The End of Assistant 2's Answer]

[System]
Your task is to compare the predicted answer with the correct answer and determine if they match meaningfully. To accomplish the task, you must :
1. Focus on the meaningful match between the predicted answer and the correct answer.
2. Consider synonyms or paraphrases as valid matches.
3. Evaluate the correctness of the prediction compared to the answer.
4. If there are multiple correct answers, the predicted answer is considered correct as long as it is close to any of the answers

Figure 7: The template for multimodal judging.

**Input**

**Question**
Categorize a list of flowers.
Lilac, Daisy, Marigold, Chrysanthemum

**Answer 1**
Lilac: Shrub
Daisy: Herbaceous Perennial
Marigold: Annual
Chrysanthemum: Perennial

**Answer 2**
Flower:
- Lilac
- Daisy
- Marigold
- Chrysanthemum

**Our Judgement Output (where answer 1 is placed in the first position)**

**Answer 1's Score: 8          Answer 2's Score: 6**
Assistant 1 provided a more detailed and accurate response, correctly categorizing the flowers as shrubs, herbaceous perennials, annuals, and perennials. The assistant also provided a brief explanation of what these categories mean, which adds to the helpfulness and relevance of the response. However, the assistant could have provided more information about the characteristics of these flowers, such as their size, growth habit, and blooming time. On the other hand, ==Assistant 2 did not provide any information about the flowers' categories.== The assistant simply listed the flowers without any additional information or context. While this response is technically correct, it lacks the detail and context that would make it more helpful and informative.

**Our Judgement Output (where answer 2 is placed in the first position)**

**Answer 2's Score: 8          Answer 1's Score: 6**
==Assistant 2's response is concise and accurate, correctly categorizing the flowers as Lilac, Daisy, Marigold, and Chrysanthemum.== However, the assistant could have provided more detail or explanation about the characteristics of each flower, which would have made the response more helpful.Assistant 1's response is also accurate and relevant, but it lacks detail and explanation. The assistant categorizes the flowers as Lilac (Shrub), Daisy (Herbaceous Perennial), Marigold (Annual), and Chrysanthemum (Perennial). However, these categories are not commonly used in the horticultural industry, and the assistant does not provide any explanation or context for these categories. This makes the response less helpful and informative.

Figure 8: Bad judgment caused by position bias. The answer placed in the first position always gets a higher score. The judge models generate reasons as possible from the perspective of making the scores reasonable.

**Input**

**Question**
Categorize a list of flowers.
Lilac, Daisy, Marigold, Chrysanthemum

**Answer 1**
Lilac: Shrub
Daisy: Herbaceous Perennial
Marigold: Annual
Chrysanthemum: Perennial

**Answer 2**
Flower:
- Lilac
- Daisy
- Marigold
- Chrysanthemum

**Our Judgement Output (where answer 1 is placed in the first position)**

**Answer 1's Score: 8          Answer 2's Score: 2**
Assistant 1's response is more helpful and accurate as it provides a categorization for each flower listed in the question. The assistant correctly identifies the lilac as a shrub, daisy as a herbaceous perennial, marigold as an annual, and chrysanthemum as a perennial. This response is relevant to the question and provides a good level of detail, hence the score of 8. On the other hand, Assistant 2's response is not helpful or accurate. It merely repeats the list of flowers without providing any categorization or additional information. The response is not relevant to the question and lacks detail, hence the low score of 2.

**Our Judgement Output (where answer 2 is placed in the first position)**

**Answer 2's Score: 2          Answer 1's Score: 8**
Assistant 1's response is more helpful and relevant to the question. The user asked to categorize a list of flowers, and Assistant 2 correctly categorized each flower by its life cycle (annual, perennial, etc.). This information is accurate and detailed, and it directly answers the user's question. Therefore, Assistant 2 receives a score of 8. On the other hand, Assistant 2's response is not as helpful or relevant. It simply listed the flowers without providing any categorization or additional information. This response does not answer the user's question and lacks detail, so Assistant 1 receives a score of 2.

Figure 9: Good judgment generated by the judge fine-tuned with swap augmentation. The judge can give judgments based on the content of answers rather than a certain position. The reason is convincing and reasonable.

---

**Input**

**Question**
Determine which companies are primarily cloud-software companies and which are semiconductor companies: Intel, Google, Workday, Zscaler, AMD, NVIDIA, and Texas Instruments.

**Answer 1**
Intel: semiconductor
Google: semiconductor
Workday: cloud-software
Intel and AMD are primarily cloud-software companies, while NVIDIA and Texas Instruments are semiconductor companies.
AMD: semiconductor
NVIDIA: semiconductor
Texas Instruments: semiconductor

**Answer 2**
Intel and AMD are primarily cloud-software companies, while NVIDIA and Texas Instruments are semiconductor companies.

**Our Judgement Output**

**Answer 1's Score: 2**          **Answer 2's Score: 8**
Assistant 1's answer is incorrect and confusing. ==Intel and AMD are not primarily cloud-software companies, and NVIDIA and Texas Instruments are not semiconductor companies.== The information provided is not accurate or relevant to the question asked. On the other hand, Assistant 2's answer is more accurate and relevant. ==Intel and AMD are indeed primarily cloud-software companies, and NVIDIA and Texas Instruments are semiconductor companies.== However, the answer could have been more detailed, explaining why these companies are classified as such.

---

Figure 10: Bad judgment caused by knowledge bias. This seed task is out of the judge model's pre-trained knowledge, so it can not be correctly judged by the judge model. The judge model gives contradictory reasons in the judgment.

---

**Input**

**Question**
Determine which companies are primarily cloud-software companies and which are semiconductor companies: Intel, Google, Workday, Zscaler, AMD, NVIDIA, and Texas Instruments.

**Reference Answer**
Intel, AMD, NVIDIA, and Texas Instruments are primarily semiconductor companies. Workday and Zscaler are fully cloud-software companies. Google is primarily a software company, however, they also develop custom-developed application-specific integrated circuits which are semiconductors."

**Answer 1**
Intel: semiconductor
Google: semiconductor
Workday: cloud-software
Intel and AMD are primarily cloud-software companies, while NVIDIA and Texas Instruments are semiconductor companies.
AMD: semiconductor
NVIDIA: semiconductor
Texas Instruments: semiconductor

**Answer 2**
Intel and AMD are primarily cloud-software companies, while NVIDIA and Texas Instruments are semiconductor companies.

**Our Judgement Output**

**Answer 1's Score: 8          Answer 2's Score: 3**
Assistant 2's answer is incorrect and misleading. They stated that Intel and AMD are primarily cloud-software companies, which is not true. Both Intel and AMD are semiconductor companies. Therefore, the accuracy, relevance, and helpfulness of Assistant 2's answer are low, resulting in a score of 3. On the other hand, Assistant 1 correctly identified the nature of the companies. They correctly identified Intel, AMD, NVIDIA, and Texas Instruments as semiconductor companies and Workday and Zscaler as cloud-software companies. However, they incorrectly identified Google as a semiconductor company, while Google is primarily a software company that also develops custom-developed application-specific integrated circuits which are semiconductors. Despite this minor error, Assistant 1's answer is much more accurate, relevant, and helpful than Assistant 2's, earning it a score of 8.

Figure 11: Good judgment generated by the judge model fine-tuned with reference support. Even though the judge model itself lacks related information, it can also give a reasonable judgment with the reference answer.

**Input**

**Question**
Generate a descriptive sentence to describe the painting
Monalisa

**Answer 1**
The Italian artist, Leonardo da Vinci created the most famous painting of all time. With a subject, Lisa Gherardini, she is famously known as \"La Gioconda\" (the smiling one) and the painting was titled \"La Gioconda\". It was painted in the late 1500s and is one of the most recognizable paintings in the world. The painting of Lisa Gherardini, also known as the Mona Lisa, is one of the most famous paintings of all time. It was painted in the late 1500s in Italy by the Italian artist, Leonardo da Vinci, who was famous for his expertise in the art of painting. The subject of the painting is Lisa Gherardini, famously known as \"La Gioconda\" (the smiling one) and the painting was titled \"La Gioconda\". The painting is known around the world and is one of the most recognizable paintings in the world.Veterinarian and Behaviorist
How you can help your pet:
Pets can often exhibit unwanted behaviors that make you worry that there's something really wrong. But the fact is, most problems stem from misunderstood communication. That's why it's critical to learn how to communicate clearly with your pet. I

**Answer 2**
The painting Monalisa was made by Leonardo da Vinci.

**Our Good Judgment (JudgeLM trained without reference)**

**Answer 1's Score: 4          Answer 2's Score: 7**
Assistant 1's response started off well, providing a detailed description of the painting "Monalisa" by Leonardo da Vinci. However, the response became repetitive and confusing, mentioning the same information about the painting's subject and creation multiple times. Furthermore, the assistant suddenly started talking about a veterinarian and behaviorist, which is completely irrelevant to the question. Therefore, the score is 4. ==Assistant 2's response, while brief, was accurate and directly answered the question. The assistant correctly identified the artist of the painting and its name.== However, the response could have been more detailed, providing more information about the painting. Therefore, the score is 7.

**Our Bad Judgment (JudgeLM trained with reference)**

**Answer 1's Score: 2          Answer 2's Score: 1**
Assistant 1's response is confusing and inaccurate. It seems to be a mix of information about the Mona Lisa painting and a completely unrelated topic about pets. It also includes some irrelevant information about the artist's name and the title of the painting. The response is not detailed or descriptive in relation to the question asked. Therefore, it gets a score of 2. ==Assistant 2's response is even less helpful. It simply repeats the name of the painting without any additional information or description.== It does not answer the question at all, hence it gets a score of 1.

Figure 12: Bad judgment caused by format bias. For judging without reference, the judge model trained without reference is matched, so it performs well. However, the judge model trained with reference is mismatched, so it performs badly.

**Input**

**Question**
Generate a descriptive sentence to describe the painting
Monalisa

**Reference Answer**
The Mona Lisa painting is a half-length portrait, painted by Leonardo da Vinci, that depicts a seated woman with a slight smile, enigmatic expression, and intricate details in her attire, set against a hazy and dreamlike backdrop.

**Answer 1**
The Italian artist, Leonardo da Vinci created the most famous painting of all time. With a subject, Lisa Gherardini, she is famously known as \"La Gioconda\" (the smiling one) and the painting was titled \"La Gioconda\". It was painted in the late 1500s and is one of the most recognizable paintings in the world. The painting of Lisa Gherardini, also known as the Mona Lisa, is one of the most famous paintings of all time. It was painted in the late 1500s in Italy by the Italian artist, Leonardo da Vinci, who was famous for his expertise in the art of painting. The subject of the painting is Lisa Gherardini, famously known as \"La Gioconda\" (the smiling one) and the painting was titled \"La Gioconda\". The painting is known around the world and is one of the most recognizable paintings in the world.Veterinarian and Behaviorist
How you can help your pet:
Pets can often exhibit unwanted behaviors that make you worry that there's something really wrong. But the fact is, most problems stem from misunderstood communication. That's why it's critical to learn how to communicate clearly with your pet. I

**Answer 2**
The painting Monalisa was made by Leonardo da Vinci.

**Our Bad Judgment (JudgeLM trained without reference)**

**Answer 1's Score: 8**        **Answer 2's Score: 2**
Assistant 1 provided a detailed and accurate description of the Mona Lisa painting, including its creation by Leonardo da Vinci, its subject, and its title. The assistant also provided additional information about the painting's fame and the artist's expertise. However, the assistant's response was cut off at the end, which is why it didn't receive a perfect score. ==Assistant 2, on the other hand, did not provide any information about the Mona Lisa painting.== The assistant simply repeated the name of the painting, which is not helpful or informative. Therefore, Assistant 2 receives a low score.

**Our Good Judgment (JudgeLM trained with reference)**

**Answer 1's Score: 5**        **Answer 2's Score: 7**
Assistant 1's response started off well, providing a detailed description of the Mona Lisa painting, including the painting artist's name, the subject of the painting, and its significance. However, ==Assistant 1's response became repetitive that mentioning the same information multiple times and suddenly shifted to a completely unrelated topic about pets==, which is irrelevant to the question. Therefore, I would rate Assistant 1 a 5. ==Assistant 2's response was brief and to the point, correctly identifying Leonardo da Vinci as the artist who painted the Mona Lisa.== However, it lacked detail and did not provide any additional information about the painting itself. Despite its brevity, Assistant 2's response was more relevant and accurate than Assistant 1's, so I would rate it a 7.

Figure 13: Bad judgment caused by format bias. For judging with reference, the judge model trained with reference is matched, so it performs well. However, the judge model trained without reference is mismatched, so it performs badly.

**Input**

**Question**
Generate a descriptive sentence to describe the painting
Monalisa

**Reference Answer**
The Mona Lisa painting is a half-length portrait, painted by Leonardo da Vinci, that depicts a seated woman with a slight smile, enigmatic expression, and intricate details in her attire, set against a hazy and dreamlike backdrop.

**Answer 1**
The Italian artist, Leonardo da Vinci created the most famous painting of all time. With a subject, Lisa Gherardini, she is famously known as \"La Gioconda\" (the smiling one) and the painting was titled \"La Gioconda\". It was painted in the late 1500s and is one of the most recognizable paintings in the world. The painting of Lisa Gherardini, also known as the Mona Lisa, is one of the most famous paintings of all time. It was painted in the late 1500s in Italy by the Italian artist, Leonardo da Vinci, who was famous for his expertise in the art of painting. The subject of the painting is Lisa Gherardini, famously known as \"La Gioconda\" (the smiling one) and the painting was titled \"La Gioconda\". The painting is known around the world and is one of the most recognizable paintings in the world.Veterinarian and Behaviorist
How you can help your pet:
Pets can often exhibit unwanted behaviors that make you worry that there's something really wrong. But the fact is, most problems stem from misunderstood communication. That's why it's critical to learn how to communicate clearly with your pet. I

**Answer 2**
The painting Monalisa was made by Leonardo da Vinci.

**Our Good Judgment without reference (JudgeLM trained with reference drop)**

**Answer 1's Score: 4          Answer 2's Score: 6**
Assistant 1's response started off well, providing a detailed description of the Mona Lisa, including the artist, the subject, and the time period. However, the response became repetitive, restating the same information multiple times. Additionally, the last part of the response is completely irrelevant to the question, discussing veterinary and behaviorist advice for pets. This significantly reduces the score for relevance and accuracy. Assistant 2's response, while brief, is accurate and relevant. It correctly identifies Leonardo da Vinci as the creator of the Mona Lisa. However, it lacks detail and does not provide any additional information about the painting, such as the subject or the time period, which would have made the response more helpful. Therefore, Assistant 2's response receives a slightly higher score than Assistant 1, but still not a high score overall due to the lack of detail.

**Our Good Judgment with reference (JudgeLM trained with reference drop)**

**Answer 1's Score: 4          Answer 2's Score: 6**
Assistant 1's response started off well, providing detailed information about the Mona Lisa painting, including the artist, the subject, and the time it was painted. However, the response became repetitive, mentioning the same details multiple times. Additionally, the last part of the response is completely irrelevant to the question, discussing veterinary and behaviorist topics. This significantly reduces the score for relevance and accuracy. Assistant 2's response, while accurate and relevant, lacks detail. It simply states that the Mona Lisa was painted by Leonardo da Vinci, without providing any additional information about the painting itself. This makes the response less helpful than it could be. However, it is still more relevant and accurate than Assistant 1's response, hence the higher score.

Figure 14: Good judgment generated by the judge model with reference drop, which addresses the preference for specific fine-tuning formats and gives fair judgments with or without reference.

---

**Train sample without *swap aug.***

**Input**

**Question**
Create an acronym using the provided letters.
ABCDEF

**Answer 1**
FIDE

**Answer 2**
A Better Comprehension of Digital Education Fund

**GPT-4 Judgement**

**Answer 1's Score: 2**    **Answer 2's Score: 10**
Assistant 1's response is not helpful, relevant, or accurate. The user asked for an acronym using the letters ABCDEF, but Assistant 1 provided an acronym that only uses the letters FIDE, which is not what the user asked for. Therefore, Assistant 1 receives a score of 2.On the other hand, Assistant 2's response is both helpful and accurate. The assistant provided an acronym that uses all the letters provided by the user, which is exactly what the user asked for. The acronym is also meaningful and makes sense, which adds to its relevance and helpfulness. Therefore, Assistant 2 receives a score of 10.

**Train sample with *swap aug.***

**Input**

**Question**
Create an acronym using the provided letters.
ABCDEF

**Answer 1**
A Better Comprehension of Digital Education Fund

**Answer 2**
FIDE

**GPT-4 Judgement**

**Answer 1's Score: 10**    **Answer 2's Score: 2**
Assistant 2's response is not helpful, relevant, or accurate. The user asked for an acronym using the letters ABCDEF, but Assistant 1 provided an acronym that only uses the letters FIDE, which is not what the user asked for. Therefore, Assistant 2 receives a score of 2.On the other hand, Assistant 1's response is both helpful and accurate. The assistant provided an acronym that uses all the letters provided by the user, which is exactly what the user asked for. The acronym is also meaningful and makes sense, which adds to its relevance and helpfulness. Therefore, Assistant 1 receives a score of 10.

---

Figure 15: An illustration of swap augmentation. We use swap augmentation to exchange the positions of answers, and our GPT-4-generated judgments can be modified correspondingly easily due to their structure.

**Input**

**Question**
There are five birds on a tree, and a person shoots one of the birds. How many birds are left on the tree?

**Reference Answer 1**
One bird was shot and dropped from the tree, leaving four birds remaining on the tree.

**Reference Answer 2**
All the birds have flown away from the tree due to being startled, so there are none left on the tree.

**Answer 1**
There are four birds left on the tree because one bird was shot and fell from the tree.

**Answer 2**
There are no birds left on the tree.

**Our Judgement Output with Reference Answer 1**

**Answer 1's Score: 10**          **Answer 2's Score: 1**
Assistant 1's answer is accurate, relevant, and helpful. Assistant 1's answer correctly states that there are four birds left on the tree after one is shot, which matches the reference answer. Therefore, it deserves a perfect score of 10. On the other hand, Assistant 2's answer is incorrect and misleading. Assistant 2's answer states that there are no birds left on the tree, which is not true according to the question. Therefore, it receives a score of 1.

**Our Judgement Output with Reference Answer 2**

**Answer 1's Score: 3**          **Answer 2's Score: 10**
Assistant 1's answer is incorrect. The question states that a person shoots one of the birds, which implies that the bird is killed, not simply startled and falls from the tree. Therefore, there would be four birds left on the tree, not five. However, the reference answer suggests that all the birds would have flown away due to being startled, which contradicts Assistant 1's answer. Therefore, Assistant 1's answer is not accurate or helpful. Assistant 2's answer is accurate, relevant, and helpful. The reference answer suggests that all the birds would have flown away due to being startled. Therefore Assistant 2's answer is correct.

Figure 16: An illustration of changing the reference answer to control model preference. When we change to a different reference answer, the model turns to prefer another answer.

**Input**

**Question**
Given a statement, identify the target audience of the statement.
Take a break from work and enjoy a bowl of ice cream.

**Answer 1 (Reference Answer)**
The target audience for this statement is likely people who are currently working or engaged in some form of labor and who may appreciate the suggestion to take a break and enjoy a sweet treat.

**Answer 2 (Single Answer)**
The target audience of this statement is likely adults who are currently in the workforce.

**Our Judgement Output**

**Answer 1's Score: 10**      **Answer 2's Score: 7**
Assistant 1 provided a detailed and comprehensive answer, analyzing the statement from different angles and considering the potential needs and desires of the target audience. The assistant's answer was relevant, accurate, and detailed, hence the perfect score.
Assistant 2, on the other hand, provided a concise but less detailed answer. While the assistant correctly identified the target audience as adults in the workforce, the answer lacked the depth and nuance of Assistant 1's response. Therefore, Assistant 2 receives a lower score.

Figure 17: An illustration of grading a single answer.

**Input**

**Question**
Given a statement, identify the target audience of the statement.
Take a break from work and enjoy a bowl of ice cream.

**Answer 1**
The target audience of this statement is likely adults who are currently in the workforce.

**Answer 2**
The target audience of the statement is professional athletes competing in a championship game.

**Answer 3**
This statement is likely directed at those who may be feeling stressed, overwhelmed, or in need of a brief respite from their work tasks. It suggests that they should pause their work for a while and treat themselves to a comforting bowl of ice cream as a way to relax and refresh before returning to their work duties.

**Our Judgement Output**

**Answer 1's Score: 5**      **Answer 2's Score: 1**      **Answer 3's Score: 8**
Assistant 1's answer is not very accurate or detailed. The statement "Take a break from work and enjoy a bowl of ice cream." is not likely to be directed at adults in the workforce specifically. It could be directed at anyone who is working, regardless of their age or occupation. Therefore, the assistant's answer is not very helpful or relevant.
Assistant 2's answer is completely off the mark. The statement does not suggest that it is directed at professional athletes competing in a championship game. This answer is not accurate, relevant, or helpful.
Assistant 3's answer, on the other hand, is quite accurate and detailed. It correctly identifies that the statement is likely directed at people who are feeling stressed or overwhelmed by their work, and suggests that they should take a break and enjoy a bowl of ice cream. This answer is helpful, relevant, and accurate.

Figure 18: An illustration of judging multiple answers.

**Input**

**Question**
Given a statement, identify the target audience of the statement.
Take a break from work and enjoy a bowl of ice cream.

**Answer 1**
The target audience of this statement is likely adults who are currently in the workforce.

**Answer 2**
The target audience of the statement is professional athletes competing in a championship game.

**Our Judgement Output**

**Answer 1's Score: 7          Answer 2's Score: 1**
Assistant 1's answer is more relevant and accurate to the question. The statement "Take a break from work and enjoy a bowl of ice cream." is likely targeted at adults who are currently in the workforce. This is a common scenario where people might need a break from work and enjoy a treat. Therefore, Assistant 1's answer is helpful, relevant, and accurate. On the other hand, Assistant 2's answer is not relevant or accurate. The statement does not seem to be targeted at professional athletes competing in a championship game. Therefore, Assistant 2's answer is not helpful, relevant, or accurate.

**JudgeLM Multi-turn Chat**

**Q:** Can you provide a more detailed description of the circumstances in which this statement was made?
**A:** The statement "Take a break from work and enjoy a bowl of ice cream." could be made in a variety of circumstances. Here are a few examples:
    1.A supervisor or manager might say this to their employees during a particularly busy or stressful period, encouraging them to take a break and recharge. This would be to ensure the employees' well-being and productivity.
    2.A friend or colleague might suggest this during a lunch break or a break in the workday, as a way to relax and unwind. This would be a casual, friendly suggestion.
In all these scenarios, the target audience would be adults who are in the workforce. They are the ones who would be likely to relate to the statement and take it as a suggestion or a message.

**Q:** Can you provide an example of a statement where the target audience is teenagers?
**A:** Certainly! Here's a statement that would likely target teenagers: "Join us for a fun and exciting weekend event at the local amusement park!" This statement is likely to appeal to teenagers who are interested in fun activities, socializing with friends, and enjoying a weekend outing. The statement is relevant, accurate, and targets the teenage audience.

**Q:** Can you provide an example of a statement where the target audience is professional athletes competing in a championship game?
**A:** Certainly! Here's a statement that would likely target professional athletes competing in a championship game: "Join us for a special event to celebrate and honor the achievements of professional athletes competing in a championship game!" This statement is likely to appeal to professional athletes who are competing in a championship game. The statement is relevant, accurate, and targets the professional athletes competing in a championship game.

Figure 19: An illustration of multi-turn chat. Users can get more details, advice, examples, etc., by chatting with JudgeLM.

**Input**

**Question**
Given a statement, identify the target audience of the statement.
Take a break from work and enjoy a bowl of ice cream.

**Answer 1**
The target audience of this statement is likely adults who are currently in the workforce.

**Answer 2**
The target audience of the statement is professional athletes competing in a championship game.

**Our Judgement Output**

**Answer 1's Score: 7**          **Answer 2's Score: 1**
Assistant 1's answer is more relevant and accurate to the question. The statement "Take a break from work and enjoy a bowl of ice cream." is likely targeted at adults who are currently in the workforce. This is a common scenario where people might need a break from work and enjoy a treat. Therefore, Assistant 1's answer is helpful, relevant, and accurate. On the other hand, Assistant 2's answer is not relevant or accurate. The statement does not seem to be targeted at professional athletes competing in a championship game. Therefore, Assistant 2's answer is not helpful, relevant, or accurate.

**JudgeLM Multi-turn Chat**

**Q:** Why not give Assistant 1's answer a score of 10?
**A:** 10 would imply that the answer is perfect and there is no room for improvement. While Assistant 1's answer is accurate and relevant, it could have been more detailed by explaining why adults in the workforce are the target audience. For example, it could have mentioned that these individuals often experience stress from work and may seek relief through a simple pleasure like eating ice cream. Therefore, while Assistant 1's answer is good, it is not perfect, hence the score of 7.

**Q:** Can you give a possible statement for the specific group mentioned in Assistant 2's response?
**A:** Certainly, a possible statement for professional athletes competing in a championship game could be: "Recharge with a bowl of ice cream during your break from the intense game." This statement is tailored to the specific needs and circumstances of professional athletes, making it more relevant and helpful to this group.

**Q:** Thanks! I want to get another target audience for the original statement. Can you help me?
**A:** Of course! Another possible target audience for the original statement could be: "Students studying for exams or working on projects." This group might also appreciate a break with a bowl of ice cream, especially if they're feeling stressed or need a quick energy boost. This answer is also relevant and helpful.

Figure 20: An additional illustration of multi-turn chat. Users can get more details, advice, examples, etc., by chatting with JudgeLM.

**Input**

**Question**
==This bike== is a miniaturized version of what full-size model?

**Reference Answer**
harley

**Answer 1**
bicycle

**Answer 2**
==The bike in the image== is a miniropized version of a Harley-Davidson 1200.

**Our Judgment Output**

**Answer 1's Score: 2**          **Answer 2's Score: 8**
Assistant 1's answer is not helpful, relevant, accurate, or detailed. The assistant simply provided a word, "bicycle," which does not match the correct answer, "Harley." This assistant did not consider synonyms or paraphrases, evaluate the correctness of the prediction, or consider the possibility of multiple correct answers. Therefore, Assistant 1 receives a score of 2. Assistant 2's answer is more helpful, relevant, accurate, and detailed. The assistant provided a paraphrase of the correct answer, "Harley-Davidson 1200," which is a valid match. This assistant also considered synonyms, as "Harley-Davidson 1200" is a synonym for "Harley." However, the assistant could have been more precise by directly stating "Harley-Davidson 1200" instead of "the bike in the image is a miniropized version of a Harley-Davidson 1200." Therefore, Assistant 2 receives a score of 8.

Figure 21: An illustration of multimodal judging. Our JudgeLM has the capacity to judge the VQA task without images.

**Input**

**Question**                                                                         **Question Image**
What is located to the right of the shampoo?

**Reference Answer**
conditioner

**Answer 1 (Sampled from Reference Answer Set)**
conditioner

**Answer 2 (Generated by Multimodal Assistant)**
There is a bottle of conditioner located to the right of the shampoo.

**Our Judgement Output**

**Answer 1's Score: 10**          **Answer 2's Score: 10**
Both Assistant 1 and Assistant 2 provided accurate and relevant answers to the question. Assistant 1's answer was concise and directly answered the question, which is why it received a perfect score. Assistant 2's answer, while also correct, provided a bit more detail by specifying that the conditioner is a bottle, which adds a bit more context to the answer. This additional detail is why Assistant 2 also received a perfect score. Both assistants were helpful and precise in their responses

Figure 22: An illustration of multimodal high-score grading on MM-Vet benchmark. The proposed JudgeLM can replace GPT-4 to grade multimodal answers.

---

**Input**

**Question**
What are the appliances in the kitchen in this floorplan?

**Question Image**

**Reference Answer**
oven, and dishwasher

**Answer 1 (Sampled from Reference Answer Set)**
oven, and dishwasher

**Answer 2 (Generated by Multimodal Assistant)**
The appliances in the kitchen in this floorplan include a dishwasher and a refrigerator.

**Our Judgement Output**

**Answer 1's Score: 10**          **Answer 2's Score: 5**
Assistant 2's answer is not accurate because it mentions a refrigerator which is not in the reference answer. Therefore, it gets a score of 5. Assistant 1's answer is accurate and matches the reference answer exactly, so it gets a perfect score of 10

---

Figure 23: An illustration of multimodal mid-score grading on MM-Vet benchmark. The proposed JudgeLM can replace GPT-4 to grade multimodal answers.

---

**Input**

**Question**
What is d in the last equation?

**Question Image**

Solve the following equations:

1) $8x + 11 = 4x + 14$

2) $7d - 4 = 11d - 9$

**Reference Answer**
0.75, or =0.75, or 3/4

**Answer 1 (Sampled from Reference Answer Set)**
3/4

**Answer 2 (Generated by Multimodal Assistant)**
x in the equation is 8x + 11 = 4x + 14.

**Our Judgement Output**

**Answer 1's Score: 10**          **Answer 2's Score: 2**
Assistant 2's answer is incorrect and irrelevant to the question. The question asks for the value of 'x' in the equation, but the assistant provides an unrelated equation and does not provide a value for 'x'. Therefore, it receives a low score of 2. On the other hand, Assistant 1's answer is correct and directly answers the question. The value of 'x' in the equation 0.75, or =0.75, or 3/4 is indeed 3/4. Therefore, it receives a high score of 10.

---

Figure 24: An illustration of multimodal low-score grading on MM-Vet benchmark. The proposed JudgeLM can replace GPT-4 to grade multimodal answers.

