# OpenReview forum: "JudgeLM: Fine-tuned Large Language Models are Scalable Judges"
_ICLR.cc/2025/Conference — ICLR 2025 Spotlight_

### Official Review · Reviewer_mQU5 · 2024-10-28

**Soundness:** 3
**Presentation:** 3
**Contribution:** 3
**Rating:** 8
**Confidence:** 4

**Summary:**

A high-quality judge LLM should ideally demonstrate the following features: strong agreement with ground truth (often human annotations, though GPT-4 judgments are acceptable in some cases), consistency to diverse input formats (resilience to various biases), inference efficiency, scalability with data and parameter size, and generalization across scenarios.
JudgeLM, as presented by the authors, offers an effective approach to optimizing these aspects:

1. **Achieving High Agreement**: The authors collect a diverse, high-quality dataset at scale through a structured pipeline (though the pipeline itself is not entirely novel).
2. **Achieving High Consistency**: To mitigate three critical biases—position, knowledge, and format biases—the authors employ three straightforward yet effective data augmentation methods: swap augmentations, reference support, and reference drop. This augmentation not only enhances consistency but also boosts agreement.
3. **Enhancing Efficiency**: The authors adopt a “grading, judging, and reasoning” pattern, as opposed to an explanation-first (CoT) approach. This method achieves a balance, trading slight reductions in agreement and consistency for increased efficiency and flexibility.
4. **Scalability**: Experimental results demonstrate JudgeLM’s scalability, as tested across varying model and data sizes. The 33B JudgeLM model even exceeds its generalist teacher, GPT-4’s agreement on a human-labeled validation set.
5. **Generalization**: JudgeLM exhibits promising generalization across various judging tasks (e.g., math problems, code generation) and diverse benchmarks (e.g., human-annotated, multimodal, and retrieval-based benchmarks).

**Strengths:**

1. **Strong Motivation**: Training a judge LLM for efficient, effective evaluation of LLMs in open-ended benchmarks is highly valuable, as it enhances privacy, consistency, and cost-effectiveness.
2. **Thorough Exploration of Key Research Questions**: The paper addresses significant questions around agreement, consistency, efficiency, scalability, and generalization for LLM-as-judge models.
3. **Solid Experiments**: A convincing ablation study supports the effectiveness of data augmentation strategies, and the “grading, judging, and reasoning” pattern retains significant agreement and consistency benefits while improving efficiency.
4. **Open-Sourced Code and Dataset**: The code and dataset are readily accessible and user-friendly, enabling further research and reproducibility.

**Weaknesses:**

1. **Assumption of GPT-4 Judgments as Ground Truth**: Although GPT-4 is a common choice for cost-effective labeling, it would be beneficial if the authors could further substantiate its reliability before using it as a standard.
2. **Suboptimal Training Query Distribution**: While the authors highlight the diversity and quality of the training data by statistics, further optimization of the ideal data distribution for training judge LLMs would add depth. What constitutes a more ideal queries distribution? For example, the distribution of user queries collected in Chatbot Arena reflects a large volume of real user queries, which is one of the reasons Chatbot Arena has become such a well-recognized benchmark. If we align the queries distribution in the training data more closely with the distribution of actual user queries, the resulting Judge LLM would evaluate samples more fairly,  drawing on the philosophy behind MixEval/MixEval-X paper.
3. **Potential Overclaiming**:
    - Bias Analysis: In Section 4, while the paper discusses “position bias” and “knowledge bias,” these concepts are well-established in prior literature. The novel contribution here lies in addressing “format bias,” so the phrase “shed light on three key biases” could overstate the novelty. It might be more precise to note that this work builds on previous discussions of position and knowledge biases while introducing and addressing format bias as a new focus.
    - Scalability Comparison: In Appendix 4, the authors mention JudgeLM’s scalability as a differentiator from PandaLM. While PandaLM also explores different model sizes (up to 70B), JudgeLM offers a more granular analysis of scalability. It would clarify the novelty here to acknowledge PandaLM’s scalability exploration but emphasize JudgeLM’s finer scalability insights.
4. **Paper Structure**: Despite the solid research content, the paper’s organization could benefit from a clearer structure, such as arranging sections by key research questions.

**Questions:**

1. In Table 3, why is JudgeLM able to judge answers in parallel, while PandaLM cannot? The authors mention certain “engineering optimizations for parallel judging”; Please provide specific details about these engineering optimizations, such as the parallel processing techniques used or any architectural changes that enable parallel judging. This would help readers better understand the efficiency advantages of JudgeLM over PandaLM.
2. In both the Abstract and the Introduction, the authors emphasize that JudgeLM achieves an agreement exceeding 90%, surpassing *human-to-human agreement*. Could you clarify this comparison? My understanding is that this primarily reflects strong internal consistency within JudgeLM’s evaluations, yet the emphasis seems to be on JudgeLM’s superior performance beyond consistency alone. If so, I suggest to provide a direct comparison between JudgeLM-to-human agreement and GPT-4-to-human agreement. This would help clarify whether JudgeLM's performance truly surpasses human-level agreement or if it's primarily a measure of internal consistency. Or if not so, feel free to correct me :)
3. In the Limitations section, the authors only mention the need to further scale up the judge dataset, which I find somewhat trivial. I recommend considering discuss more improvements to the dataset, such as refining the data distribution to real user quires. This could enhance JudgeLM’s performance in areas where the current data may lack diversity or balance. Please refer to my comments in the weaknesses section for further insights.

I will increase the score if all these concerns are addressed.

---

> ### Author Response · Authors · 2024-11-23
> **To reviewer mQU5/#4 (1/2)**
>
> **Concern 1:** Assumption of GPT-4 Judgments as Ground Truth: Although GPT-4 is a common choice for cost-effective labeling, it would be beneficial if the authors could further substantiate its reliability before using it as a standard.
>
> **Response 1:** Thank you for raising this important question. We conduct more experiments to further evaluate the reliability of GPT-4's annotation. In this experiment setting, we set annotations provided by Human 1 in our benchmark as ground truth. Then we calculate the agreement of JudgeLM-33B, GPT-4, and Human 2 as shown in Table R3.
>
> These results demonstrate that GPT-4 achieves higher agreement with the ground truth compared to Human 2. llm-as-a-judge [1] shows similar results, which means GPT-4 teacher's judgments can align with human evaluators closely and can serve as a human-like teacher judge to provide high-quality and reliable judgments. We have added this comparison to the revised paper appendix.
>
> |                        | JudgeLM-33B | GPT-4 | Human 2 |
> |------------------------|:-----------:|:-----:|:-------:|
> | Ground Truth (Human 1) |    90.72    | 84.48 |  79.82  |
>
> Table R3. Agreement evaluation of JudgeLM-33B, GPT-4, and Human, with another Human's judgments as ground truth.
>
>
>
> [1] Zheng L, Chiang W L, Sheng Y, et al. Judging llm-as-a-judge with mt-bench and chatbot arena.
>
>
>
> **Concern 2:** Suboptimal Training Query Distribution: While the authors highlight the diversity and quality of the training data by statistics, further optimization of the ideal data distribution for training judge LLMs would add depth. What constitutes a more ideal queries distribution? For example, the distribution of user queries collected in Chatbot Arena reflects a large volume of real user queries, which is one of the reasons Chatbot Arena has become such a well-recognized benchmark. If we align the queries distribution in the training data more closely with the distribution of actual user queries, the resulting Judge LLM would evaluate samples more fairly, drawing on the philosophy behind MixEval/MixEval-X paper.
>
> **Response 2:** Thank you for your insightful suggestions about training query distribution. We agree that the distribution of training queries is a key factor in shaping the evaluation capabilities of Judge LLMs. While our current dataset was designed to ensure task diversity, MixEval[2] and MixEval-X[3] give us more insight, i.e., aligning the query distribution more closely with real-world user queries could further enhance the model’s fairness and relevance. We leave it as a future work and include this plan in the revised manuscript.
>
>
>
> [2] Ni J, Xue F, Yue X, et al. MixEval: Deriving Wisdom of the Crowd from LLM Benchmark Mixtures.
>
> [3] Ni J, Song Y, Ghosal D, et al. MixEval-X: Any-to-Any Evaluations from Real-World Data Mixtures.
>
>
>
>
>
> **Concern 3:** Potential Overclaiming:
>
> - Bias Analysis: In Section 4, while the paper discusses “position bias” and “knowledge bias,” these concepts are well-established in prior literature. The novel contribution here lies in addressing “format bias,” so the phrase “shed light on three key biases” could overstate the novelty. It might be more precise to note that this work builds on previous discussions of position and knowledge biases while introducing and addressing format bias as a new focus.
>
> - Scalability Comparison: In Appendix 4, the authors mention JudgeLM’s scalability as a differentiator from PandaLM. While PandaLM also explores different model sizes (up to 70B), JudgeLM offers a more granular analysis of scalability. It would clarify the novelty here to acknowledge PandaLM’s scalability exploration but emphasize JudgeLM’s finer scalability insights.
>
>
> **Response 3:** Thank you for your feedback regarding the potential overclaiming in our discussion of biases and scalability. We recognize the importance of accurately positioning our contributions relative to prior work and have revised the relevant sections to better reflect this.
>
> **Bias** **Analysis**: We have revised the manuscript and changed the phrase "shed light on three key biases in fine-tuning LLMs as judges" to "analyze two key biases and introduce a new format bias in fine-tuning LLMs as judges"
>
> **Scalability Comparison**: We will revise Appendix 4 to acknowledge PandaLM’s contributions while emphasizing the finer scalability insights provided by JudgeLM.
>
> These revisions aim to more accurately position our work within the existing research landscape and highlight our unique contributions.

---

> ### Author Response · Authors · 2024-11-23
> **To reviewer mQU5/#4 (2/2)**
>
> **Concern 4:** Paper Structure: Despite the solid research content, the paper’s organization could benefit from a clearer structure, such as arranging sections by key research questions.
>
> **Response 4:** Thank you for your thoughtful feedback regarding the paper's structure. We appreciate your suggestion to organize the sections around key research questions, as this could enhance clarity in some contexts.
>
> However, our current structure is designed to guide readers through our research process step by step, starting with the identification of key bias, followed by our proposed methods, and concluding with experimental results. This approach ensures a logical flow that aligns with the objectives of our study and the typical expectations of our target audience.
>
> While we feel that the existing structure effectively conveys the content and findings, we value your perspective and will consider adopting a research-question-based organization in our future extension work.
>
>
>
> **Concern 5:** In Table 3, why is JudgeLM able to judge answers in parallel, while PandaLM cannot? The authors mention certain “engineering optimizations for parallel judging”; Please provide specific details about these engineering optimizations, such as the parallel processing techniques used or any architectural changes that enable parallel judging. This would help readers better understand the efficiency advantages of JudgeLM over PandaLM.
>
> **Response 5:** Thank you for your thoughtful comment regarding the parallel judging capabilities of JudgeLM. We will provide detailed information about the engineering optimizations of JudgeLM.
>
> **Engineering Optimizations**: JudgeLM employs ray, a distributed computing framework, to distribute inference tasks across up to 8 GPUs on a single machine. Each GPU processes a subset of samples in parallel, and the results are subsequently aggregated. This implementation provides an approximate 8× improvement in inference throughput compared to PandaLM, whose codebase uses a naive single-GPU inference pipeline.
>
> Furthermore, JudgeLM incorporates a “grading, judging, and reasoning” modeling pipeline. When the reasoning phase is unnecessary, the model skips this step, resulting in an additional 16.65× speedup for specific tasks.
>
>
>
> **Concern 6:** In both the Abstract and the Introduction, the authors emphasize that JudgeLM achieves an agreement exceeding 90%, surpassing human-to-human agreement. Could you clarify this comparison? My understanding is that this primarily reflects strong internal consistency within JudgeLM’s evaluations, yet the emphasis seems to be on JudgeLM’s superior performance beyond consistency alone. If so, I suggest to provide a direct comparison between JudgeLM-to-human agreement and GPT-4-to-human agreement. This would help clarify whether JudgeLM's performance truly surpasses human-level agreement or if it's primarily a measure of internal consistency. Or if not so, feel free to correct me :)
>
> **Response 6:** Thank you for your insightful feedback. We conduct more experiments to further clarify the comparison between JudgeLM and human-to-human agreement. In this experiment setting, we set annotations provided by Human 1 in our benchmark as ground truth. Then we calculate the agreement of JudgeLM-33B, GPT-4, and Human 2 as shown in Table R3.
>
> These results demonstrate that JudgeLM not only achieves higher agreement with the ground truth compared to GPT-4 but also exceeds the agreement between human evaluators (Human 1 vs. Human 2). JudgeLM's performance indicates its ability to provide judgments that align closely with high-quality human annotations. llm-as-a-judge[1] shows similar results, which means LLM judges' judgments can align with human evaluators closely and can help humans make better judgments.
>
> Regarding your suggestion to provide a direct comparison between JudgeLM-to-human and GPT-4-to-human agreement, Table R3 directly addresses this by showing JudgeLM achieving a significantly higher agreement rate (90.72%) compared to GPT-4 (84.48%). This comparison highlights the effectiveness and reliability of JudgeLM as an evaluation framework. We have added this comparison to the revised paper appendix.
>
>
>
> **Concern 7:** In the Limitations section, the authors only mention the need to further scale up the judge dataset, which I find somewhat trivial. I recommend considering discuss more improvements to the dataset, such as refining the data distribution to real user quires. This could enhance JudgeLM’s performance in areas where the current data may lack diversity or balance. Please refer to my comments in the weaknesses section for further insights.
>
> **Response 7:** Thank you for highlighting the importance of data distribution. As mentioned in Response 2, we leave more exploration of data distribution as future work and include this plan in the revised manuscript.

---

> > ### Comment · Reviewer_mQU5 · 2024-11-23
> >
> > Thank you for explaining everything so clearly. I learned a lot from your work, and I like how solid the paper is. My suggestion about the structure is just an idea for the future, not a request to change the current paper. I have updated my score to 8.

---

> > > ### Author Response · Authors · 2024-11-27
> > >
> > > Thank you so much for your kind words and strong support! We will keep moving to make JudgeLM and the following projects better and better.

---

### Official Review · Reviewer_GP51 · 2024-11-01

**Soundness:** 3
**Presentation:** 3
**Contribution:** 2
**Rating:** 8
**Confidence:** 4

**Summary:**

The paper introduces JudgeLM, a scalable framework for evaluating LLMs by fine-tuning models as judges, using a dataset of LLM-generated answers and GPT-4 judgments. This framework addresses evaluation biases such as position and format biases through techniques like swap augmentation and reference drop. The authors show that JudgeLM achieves high alignment with the GPT-4 judge and is more efficient than comparable methods.

**Strengths:**

- JudgeLM’s ability to process up to 5K samples in 3 minutes on an 8-GPU system is impressive, and it supports multiple use cases including single-answer and multimodal evaluations.
- The swap augmentation and reference-based adjustments offer a nice way to mitigate biases that impact LLM judgments, contributing to more reliable scoring across scenarios.
- The JudgeLM dataset, with its human-verified judgments and optional reference answers, is a notable contribution.

**Weaknesses:**

- The model's robustness under varied task complexities or unseen domains is not extensively tested (e.g. math/coding/reasoning task). Additional benchmarks or diverse human annotations would reinforce its generalizability.
- The study acknowledges that scaling up the judge dataset is costly and currently relies on GPT-4 outputs. Exploring alternative sources or synthetic judgment data could be beneficial.

**Questions:**

- Can JudgeLM’s performance be reliably extended to evaluate answers in different languages or domain-specific contexts (e.g., math, coding, legal or medical)?
- How does the model handle cases where the reference answer may introduce bias rather than mitigate it?

---

> ### Author Response · Authors · 2024-11-23
> **To reviewer GP51/#3 (1/2)**
>
> **Concern 1:** The model's robustness under varied task complexities or unseen domains is not extensively tested (e.g. math/coding/reasoning task). Additional benchmarks or diverse human annotations would reinforce its generalizability.
>
> **Response 1:** Thank you for raising this important question regarding JudgeLM’s robustness under varied task complexities and unseen domains. We agree that thorough evaluation in these areas is essential to understanding its generalizability. To this end, we have conducted extensive experiments across a diverse range of tasks and benchmarks, as detailed in Appendix A.3.
>
> **Domain-specific Tasks**: As shown in Table 9 and Table 11, the JudgeLM benchmark includes diverse domains such as math, coding, planning, and science. Specifically, JudgeLM achieves **[86.45% agreement]** on math tasks, **[88.08% agreement]** on coding tasks, and **[88.41% agreement]** on common-sense tasks, demonstrating strong performance across these complex domains.
>
> **Unseen Benchmarks**: To further evaluate its generalizability, we tested JudgeLM on several challenging benchmarks:
>
> - **Multimodal tasks** (Table 12): Achieving **[91.74% agreement]**, outperforming the 0-shot GPT-4 and GPT-3.5 judges.
>
> - **Toxic conversation evaluation** (Table 13): Scoring **[89.66% accuracy]**, achieving superior precision and comparable accuracy to the specialist model, i.e., OpenAI Moderation.
>
> - **Retrieval-based tasks** (Table 14): Achieving **[over 85.5% agreement]** with over 400 words, demonstrating adaptability to retrieval-based tasks.
>
> These results confirm JudgeLM’s robustness in handling diverse and unseen tasks. Furthermore, JudgeLM can further generalize to multiple-format judging, single-answer grading, reward model, and multi-turn chat about judgment as presented in Appendix A.3. These results collectively highlight the generalization of JudgeLM, not only on the PandaLM test set but also across diverse out-of-distribution scenarios.
>
>
>
> **Concern 2:** The study acknowledges that scaling up the judge dataset is costly and currently relies on GPT-4 outputs. Exploring alternative sources or synthetic judgment data could be beneficial.
>
> **Response 2:** Thank you for your thoughtful comment on the dataset’s reliance on GPT-4. We totally agree with your suggestion to incorporate more diverse teacher models for making JudgeLM's judgments more adaptable and introduce synthetic judge data to reduce the dataset cost.
>
> As mentioned in Appendix A.1, we recognize the importance of diversifying the dataset and have identified incorporating high-quality annotations from models such as Claude and Qwen [1, 2] as a promising direction for future work. Different teacher models exhibit complementary strengths and weaknesses, which could minimize over-reliance on any single model and improve the adaptability of JudgeLM’s judgments. Expanding the dataset to include annotations from multiple teacher models remains a key priority for our future work.
>
> On the other hand, motivated by the recent work [3], which shows a promising way to finetune more advanced models with a weak supervisor: OpenAI researchers find that when they naively finetune strong pretrained models on labels generated by a weak model, the fine-tuned models consistently perform better than the weak supervisors, a phenomenon they call weak-to-strong generalization. The larger-size JudgeLM model can also serve as a weak supervisor to generate labels to fine-tune strong pretrained models. We also leave it in future work.
>
>
>
> [1] Bai J, Bai S, Chu Y, et al. Qwen technical report.
>
> [2] Yang A, Yang B, Hui B, et al. Qwen2 technical report.
>
> [3] Burns C, Izmailov P, Kirchner J H, et al. Weak-to-strong generalization: Eliciting strong capabilities with weak supervision.

---

> ### Author Response · Authors · 2024-11-23
> **To reviewer GP51/#3 (2/2)**
>
> **Concern 3:** Can JudgeLM’s performance be reliably extended to evaluate answers in different languages or domain-specific contexts (e.g., math, coding, legal or medical)?
>
> **Response 3:** Thanks for this important question. We have evaluated the generalization ability of JudgeLM on Domain-specific Tasks. As shown in Table 9 and Table 11, the JudgeLM val set includes diverse domains such as math, coding, planning, and science. Specifically, JudgeLM achieves **[86.45% agreement]** on math tasks, **[88.08% agreement]** on coding tasks, and **[88.41% agreement]** on common-sense tasks, demonstrating strong performance across these complex domains. We further evaluate JudgeLM on legal (also denoted as law), and medical (also denoted as health) tasks, and JudgeLM achieves 87.74% agreement and 88.12% agreement, respectively.
>
> The support for different languages mainly relies on the base model. We conduct experiments with Vicuna base models and it shows superior performance in English. For judging in other languages, we suggest using a multi-language-pretrained base model like Qwen 2.5 [4], and incorporating multi-language-translation on the JudgeLM train set as a data-augmentation method. We leave it as future work.
>
>
>
> [4] Team Q. Qwen2. 5: A party of foundation models, https://qwenlm.github.io/blog/qwen2.5
>
>
>
> **Concern 4:** How does the model handle cases where the reference answer may introduce bias rather than mitigate it?
>
> **Response 4:** Thank you for raising this important concern. We acknowledge that reference answers, while intended to guide consistent and reliable judgments, may occasionally introduce bias if they are incomplete, subjective, or reflect a specific perspective. To address this, we have implemented the following quality control measures during dataset construction:
>
> **Automated Filtering**: Reference answers are filtered using automated scripts to remove incorrect, irrelevant, or harmful content.
>
> **Human Validation**: The judgments in the dataset are manually checked and refined by authors to ensure accuracy, neutrality, and completeness.
>
> These efforts aim to further reduce the potential for bias in reference answers and enhance JudgeLM’s robustness and reliability.

---

> ### Author Response · Authors · 2024-11-30
> **Thanks for Your Positive Comments**
>
> Thank you very much for your thoughtful review and positive comments. We sincerely appreciate your time and the recognition of our work. We hope our responses have addressed all your concerns, and we are happy to provide any further clarifications if needed.

---

### Official Review · Reviewer_1FsA · 2024-11-02

**Soundness:** 3
**Presentation:** 3
**Contribution:** 3
**Rating:** 6
**Confidence:** 3

**Summary:**

The paper proposes a scalable method for evaluating LLMs in open-ended tasks. It leverages a large, diverse dataset with tasks, LLM-generated answers, and GPT-4 judgments to fine-tune models that assess other models effectively. To mitigate biases such as position, knowledge, and format biases, the authors introduce techniques like swap augmentation, reference support, and reference drop. JudgeLM achieves high agreement rates with GPT-4, surpassing human-level consistency, and can judge multiple formats efficiently.

**Strengths:**

The paper demonstrates a novel approach to scaling LLM evaluation by fine-tuning models as judges, creating a comprehensive system that addresses biases inherent in model evaluations. This creative combination of fine-tuning techniques with practical augmentation methods (swap, reference support, reference drop) removes limitations from prior works that struggled with consistent, scalable evaluation in open-ended tasks.

The quality of the work is solid, backed by a large-scale, carefully curated dataset, including GPT-4 judgments and human validation, which strengthens the empirical basis of the results. In terms of clarity, the paper effectively communicates its methodology and contributions.

Given the increasing role of LLMs across various fields, there is pressing need for scalable, unbiased evaluation frameworks. JudgeLM’s seems a valuable tool in AI evaluation, with potential impact on future benchmarks and research in LLM development.

**Weaknesses:**

The improvements seem The dataset, although extensive, primarily relies on GPT-4 for initial judgments, which may inadvertently transfer GPT-4’s specific limitations to JudgeLM. A more diverse range of teacher models such as Claude could minimize over-reliance on any single model’s limitations, making JudgeLM’s judgments more adaptable.

**Questions:**

Given that JudgeLM reportedly surpasses human-to-human agreement, how does it handle cases where human judgments may rely on subjective or context-dependent insights? Could the authors discuss potential scenarios where JudgeLM might still fall short of human evaluators in nuanced judgments, and possibly include or consider adding complex tasks to explore this?

---

> ### Author Response · Authors · 2024-11-23
> **To reviewer 1FsA/#2**
>
> **Concern 1:** The improvements seem The dataset, although extensive, primarily relies on GPT-4 for initial judgments, which may inadvertently transfer GPT-4’s specific limitations to JudgeLM. A more diverse range of teacher models such as Claude could minimize over-reliance on any single model’s limitations, making JudgeLM’s judgments more adaptable.
>
> **Response 1:** Thank you for your thoughtful comment on the dataset’s reliance on GPT-4 for initial judgments. We agree with your suggestion to incorporate more diverse teacher models for making JudgeLMj's judgments more adaptable.
>
> As mentioned in Appendix A.1, we recognize the importance of diversifying the dataset and have identified incorporating high-quality annotations from models such as Claude and Qwen [1, 2] as a promising direction for future work. Different teacher models exhibit complementary strengths and weaknesses, which could minimize over-reliance on any single model and improve the adaptability of JudgeLM’s judgments. Expanding the dataset to include annotations from multiple teacher models remains a key priority for our future work.
>
>
>
> [1] Bai J, Bai S, Chu Y, et al. Qwen technical report.
>
> [2] Yang A, Yang B, Hui B, et al. Qwen2 technical report.
>
>
>
> **Concern 2:** Given that JudgeLM reportedly surpasses human-to-human agreement, how does it handle cases where human judgments may rely on subjective or context-dependent insights?
>
> **Response 2:** Thank you for raising this important concern. We acknowledge that subjective and context-dependent judging tasks pose unique challenges for all judge models, as they often require judgment that can vary across contexts or depend on specific individual preferences. LLM judges, like human judges, may exhibit their own preferences when handling such cases.[3]
>
> To address this issue, we proposed the Judge with a reference method. By introducing an explicit reference, JudgeLM aligns its judgments with the defined preferences outlined by the reference, thereby reducing ambiguity and inconsistency. As shown in Figure 16, JudgeLM leverages the reference to guide its decision-making process, ensuring that subjective preferences or contextual ambiguities are minimized.
>
>
>
> [3] Li J, Zhou F, Sun S, et al. Dissecting Human and LLM Preferences.
>
>
>
> **Concern 3:** Could the authors discuss potential scenarios where JudgeLM might still fall short of human evaluators in nuanced judgments, and possibly include or consider adding complex tasks to explore this?
>
> **Response 3:** Thank you for pointing out the importance of exploring complex tasks for JudgeLM. We agree that current limitations in its architecture may prevent it from fully addressing certain challenges, such as those requiring multi-modal judging.
>
> One notable limitation is that JudgeLM, being based on a purely text-based LLM, lacks native visual perception capabilities. This limitation is evident in its performance on the Multimodal Judging Benchmark (as shown in Table 12), where JudgeLM achieves 91.74% agreement with human evaluators. While this performance is strong, it highlights that JudgeLM relies solely on textual references to interpret image-related content, which may limit its ability to make fully informed judgments in multi-modal contexts.
>
> To address this issue, we propose a future direction of developing a multimodal Judge by incorporating multi-modal models as the base model for JudgeLM. This approach would enable JudgeLM to natively process and evaluate multi-modal inputs, such as text-image pairs, thereby enhancing its capability to handle complex multi-modal tasks. We believe this direction has the potential to significantly improve the robustness and adaptability of JudgeLM in diverse real-world applications.

---

> ### Author Response · Authors · 2024-11-30
> **Anticipating Your Participation as Reviewer-Author Discussion Deadline Approaching**
>
> We express our sincere gratitude to the reviewer for dedicating time to review our paper. We have provided comprehensive responses to all the concerns. As the discussion deadline looms within 3 days, we would like to inquire if our responses have adequately addressed your questions. We are more than willing to address any concerns and ensure a comprehensive resolution. Thank you for your time and consideration.

---

### Official Review · Reviewer_TdZL · 2024-11-02

**Soundness:** 4
**Presentation:** 4
**Contribution:** 3
**Rating:** 8
**Confidence:** 4

**Summary:**

This paper targets evaluating and building LLM's specifically for judging answer correctness on open-ended tasks. To do this, they construct a dataset which consists of llm answers across a variety of tasks along with GPT-4 generated judgements used as a ground truth (ground truth references responses are sometimes supplied). This dataset is used to train smaller LLMs to provide judgements with comparable accuracy to SOTA LLMs. They also discuss the biases that result from the LLM judge finetuning (position bias, knowledge bias, format bias) and propose methods to address them.

**Strengths:**

- The problem setting of building cheaper, scalable LLM judges is an important problem now that LLM-driven evaluation is becoming standard, and the provided benchmark will be incredibly valuable to the community
- This paper is easy to follow and provides a comprehensive analysis comparing JudgeLM to existing LLM judges on both accuracy and efficiency
- Improvements over previous LLM judges is impressive and provides a promising alternative to expensive closed source model judges

**Weaknesses:**

- Knowledge bias is not a bias but rather a limitation of the LLM, and the proposed solution to this seems to be providing that knowledge via a reference (AKA making the out of distribution task in distribution). While I am not opposed to this solution, I would argue this is not a bias addressed but rather a universally known failure case of the model that the authors try to mitigate by training on more knowledge via reference examples - a universally known solution to address model knowledge gaps.
- While the validation set was manually checked and corrected by the authors, it does still rely on GPT generated outputs. This provides somewhat of an unfair evaluation as JudgeLM is trained on GPT generated judgements as well. Even with the human validation, there is a reasonable chance that if this dataset where annotated by a different LLM and produced different judgements, humans checking responses would also consider them reasonable. An unbiased way of annotating is for humans to provide judgements *without* knowing what the GPT judgement is. If the agreement between humans and the GPT judgements are similar, than I would consider this evaluation relatively fair across all judge models.
- The authors did not provide clear evidence that this model is able to maintain good performance across tasks not in the training set. I suspect that the comparison to the PandaLM test set is showing this to some extent, but I did not see any prose on *how* these two datasets differ. What tasks are seen in PandaLM that arent seen in the JudgeLM dataset? If the authors can show that the task distribution is significantly different from the training set I would be satisfied

**Questions:**

- did you notice any favoring of GPT-4 answers by the JudgeLM?
- What tasks are seen in PandaLM that arent seen in the JudgeLM dataset?

---

> ### Author Response · Authors · 2024-11-23
> **To reviewer TdZL/#1 (1/3)**
>
> **Concern 1:** Knowledge bias is not a bias but rather a limitation of the LLM, and the proposed solution to this seems to be providing that knowledge via a reference (AKA making the out of distribution task in distribution). While I am not opposed to this solution, I would argue this is not a bias addressed but rather a universally known failure case of the model that the authors try to mitigate by training on more knowledge via reference examples - a universally known solution to address model knowledge gaps.
>
> **Response 1:** We appreciate the reviewer's insightful feedback and discussion. We agree that "knowledge bias" stems from the knowledge gaps between judge samples and the incomplete domain-specific knowledge encoded in pre-trained models. In such cases, LLM judges may rely on pre-trained, yet undesirable, out-of-date, or inaccurate knowledge to resolve the judge samples, leading to wrong judgments. This tendency to judge out-of-distribution tasks with inappropriate pre-trained knowledge is what we define as "knowledge bias."
>
> Regarding the proposed solution, while we acknowledge that using references to address knowledge gaps is a well-established strategy, our work specifically focuses on systematically tackling this issue in out-of-distribution tasks and rigorously quantifying the resulting improvements. As demonstrated in the ablation study (Table 6), incorporating reference support leads to significant performance gains, underscoring the effectiveness and contribution of our approach.
>
>
> **Concern 2:** While the validation set was manually checked and corrected by the authors, it does still rely on GPT generated outputs. This provides somewhat of an unfair evaluation as JudgeLM is trained on GPT generated judgements as well. Even with the human validation, there is a reasonable chance that if this dataset where annotated by a different LLM and produced different judgements, humans checking responses would also consider them reasonable. An unbiased way of annotating is for humans to provide judgements without knowing what the GPT judgement is. If the agreement between humans and the GPT judgements are similar, than I would consider this evaluation relatively fair across all judge models.
>
> **Response 2:** Thank you for highlighting this important question. We would like to clarify our current data annotation process to address your points. The generation process involves the following steps:
>
> 1. **Initial annotation**: GPT-4 provides a preliminary set of judgments.
>
> 2. **Human re-annotation**: Authors independently provide a simple judgment (“Answer 1 wins,” “Answer 2 wins,” or “Tie”) without GPT-4’s annotations as priors.
>
> 3. **Cross-validation and refinement**: We compare the independent human judgments with GPT-4’s annotations, thoroughly checking judge results, judge scores, and the quality of reasoning to ensure accuracy and reliability.
>
> Therefore, our process already aligns closely with the “human annotation without GPT exposure” method you suggested. We believe this independent human evaluation mitigates potential bias from the initial GPT annotations. Additionally, the cross-validation step serves as a safeguard to further improve the quality and reliability of the annotations.
>
> To strengthen our manuscript and address your concerns, we have explicitly clarified these steps in the appendix.

---

> ### Author Response · Authors · 2024-11-23
> **To reviewer TdZL/#1 (2/3)**
>
> **Concern 3:** The authors did not provide clear evidence that this model is able to maintain good performance across tasks not in the training set. I suspect that the comparison to the PandaLM test set is showing this to some extent, but I did not see any prose on how these two datasets differ. What tasks are seen in PandaLM that arent seen in the JudgeLM dataset? If the authors can show that the task distribution is significantly different from the training set I would be satisfied
>
> **Response 3:** Thank you for raising this important concern. To address your point, we provide evidence from both the PandaLM test set and the proposed JudgeLM benchmark to demonstrate the generalization ability of JudgeLM:
>
>
> |                       | count | percentage | delta percentage |           | count | percentage | delta percentage |     |       |        |
> | :-------------------: | :---: | :--------: | :--------------: | :-------: | :---: | :--------: | :--------------: | --- | ----- | ------ |
> |     **business**      |  87   |   8.71%    |      8.71%       |  writing  |  81   |   8.11%    |      -4.39%      |     |       |        |
> |      **fact-QA**      |  70   |   7.01%    |      7.01%       | planning  |  57   |   5.71%    |      -0.47%      |     |       |        |
> |    **summarizing**    |  64   |   6.41%    |      6.41%       | roleplay  |  57   |   5.71%    |      4.17%       |     |       |        |
> |    **linguistics**    |  45   |   4.50%    |      4.50%       |  coding   |  54   |   5.41%    |      1.39%       |     |       |        |
> |      **emotion**      |  45   |   4.50%    |      4.50%       |    art    |  44   |   4.40%    |      -2.30%      |     |       |        |
> | **entity-processing** |  42   |   4.20%    |      4.20%       |  finance  |  40   |   4.00%    |      1.16%       |     |       |        |
> |      **explain**      |  41   |   4.10%    |      4.10%       |  culture  |  38   |   3.80%    |      -0.86%      |     |       |        |
> |     **retrieval**     |  40   |   4.00%    |      4.00%       |   math    |  37   |   3.70%    |      -1.30%      |     |       |        |
> |     **document**      |  30   |   3.00%    |      3.00%       | geography |  12   |   1.20%    |      -2.78%      |     |       |        |
> |       **chat**        |  26   |   2.60%    |    2.60%    |      others      | 5   | 0.50% | -0.76% |
> |    recommendation     |  84   |   8.41%    |      -1.23%      |   total   |  999  |  100.00%   |                  |     |       |        |
>
> Table R1. Distribution of question categories in PandaLM test set. The delta percentage represents the percentage difference compared to the JudgeLM benchmark. We **bolded** categories that appear in the PandaLM test set but don't exist in the JudgeLM benchmark.
>
> **Task Distribution Differences**: An analysis of task distributions in Table R1 shows significant differences between the PandaLM test set and the JudgeLM benchmark. For example, business, fact-QA, summarizing, linguistics, emotion, entity-processing, explain, retrieval, document, and chat are well-represented in PandaLM but absent in JudgeLM, while writing and roleplay show a significant delta percentage (over 4%). This confirms that the PandaLM test set includes 49% unseen task samples that are out of distribution for JudgeLM. We have added this comparison to the revised paper appendix.
>
> As detailed in Appendix A.3, JudgeLM has been further tested on **additional unseen benchmarks**, including:
>
> - **Multimodal tasks** (Table 12): Achieving **[91.74% agreement]**, outperforming the 0-shot GPT-4 and GPT-3.5 judges.
>
> - **Toxic conversation evaluation** (Table 13): Scoring **[89.66% accuracy]**, achieving superior precision and comparable accuracy to the specialist model, i.e., OpenAI Moderation.
>
> - **Retrieval-based tasks** (Table 14): Achieving **[over 85.5% agreement]** with over 400 words, demonstrating adaptability to retrieval-based tasks.
>
> Across these benchmarks, JudgeLM demonstrates strong performance, often matching or exceeding other strong methods, despite the tasks being entirely unseen during training. Moreover, JudgeLM can further generalize to multiple-format judging, single-answer grading, reward model, and multi-turn chat about judgment as presented in Appendix A.3.
>
> These results collectively highlight the generalization of JudgeLM, not only on the PandaLM test set but also across diverse out-of-distribution scenarios.

---

> ### Author Response · Authors · 2024-11-23
> **To reviewer TdZL/#1 (3/3)**
>
> **Concern 4:** did you notice any favoring of GPT-4 answers by the JudgeLM?
>
> **Response 4:** Thank you for discussing this interesting phenomenon. We carefully conducted experiments between GPT-4 and GPT-3.5 to analyze the potential favoring of GPT-4 answers by JudgeLM. As shown in Table R2, GPT-4 wins 59.0% of pairwise comparisons, with only 16.2% of wins for GPT-3.5. We further conducted a sampling analysis to identify the following contributing factors:
>
> 1. **Alignment with Prompt Requirements**: As shown in the judging templates (Fig. 3 to Fig. 7), JudgeLM evaluates responses based on structured criteria, such as relevance and level of detail. Our analysis shows that GPT-4 answers tend to align better with these requirements compared to GPT-3.5 answers.
>
> 2. **Inherent Quality** **Difference**: A sampling analysis of 500 cases (approximately 16.9% of the GPT-4 win cases) showed that in 482 instances (96.4%), GPT-4 answers were more accurate, detailed, and contextually relevant than GPT-3.5 answers. This analysis underscores that GPT-4 win cases are overwhelmingly attributed to the higher answer quality, rather than the preference from JudgeLM.
>
> Moreover, the sample analysis shows that when the quality of the answers to GPT-4 and GPT-3.5 are similar, JudgeLM tends to give similar scores, resulting in a Tie. These results indicate that GPT-4 answers win more due to their better relevance, high level of detail, and high quality. We have included this analysis and discussion in the revised manuscript for clarity.
>
> |                   | GPT4 win |   Tie  | GPT3.5 win |
> |-------------------|:--------:|:------:|:----------:|
> | GPT4 v.s. GPT-3.5 |  59.00%  | 24.80% |   16.20%   |
>
> Table R2. Quantitative comparison results between GPT-4 and GPT-3.5.
>
> **Concern 5:** What tasks are seen in PandaLM that arent seen in the JudgeLM dataset?
>
> **Response 5:** Thanks for this kind question. As mentioned in Response 3, business, fact-QA, summarizing, linguistics, emotion, entity-processing, explain, retrieval, document, and chat are well-represented in PandaLM but absent in JudgeLM.

---

> ### Comment · Reviewer_TdZL · 2024-11-26
> **Incredible Response**
>
> The authors have gone above and beyond to address my concerns. I will happily keep my accept rating and encourage reviewer 1FsA to respond, as many of their concerns seem to have been addressed.

---

> > ### Author Response · Authors · 2024-11-27
> >
> > Thank you so much for your kind words and strong support! We truly appreciate the insightful and thoughtful feedback provided by all the reviewers, and it is our pleasure to address their concerns. We remain committed to continuously improving JudgeLM and our future projects.

---

### Author Response · Authors · 2024-11-23
**Global Response**

We sincerely appreciate all reviewers for their time and efforts in reviewing our paper. We are grateful to the reviewers for the encouraging positive comments, such as "**valuable**" (reviewer TdZL/#1, 1FsA/#2, and mQU5/#4), "**solid**" (reviewer 1FsA/#2, and mQU5/#4), "**comprehensive**" (reviewer TdZL/#1, and 1FsA/#2), "**impressive**" (reviewer TdZL/#1, and GP51/#3). We believe that a **strong**, **versatile**, and **accessible** LLM judge, as well as the evaluation for LLM, is crucial for the field's advancement. It represents a significant step in harnessing the open-ended capabilities of open-sourced LLMs to judge themselves. We have diligently addressed all of the reviewers' concerns in the corresponding responses.

---

### Meta-Review · Area_Chair_jpPf · 2024-12-19

**Metareview:**

This paper introduces resources to enable training of JudgeLM, a language model trained specifically to behave as an automatic evaluator. A large-scale dataset to train judge models is proposed, covering diverse seed tasks. The data also includes generated answers and detailed responses generated by GPT-4. A training recipe is also contributed. Since multiple answers are obtained for each given query during data generation, training follows the instruction tuning framework. Authors evaluate trained models and observe a high agreement rate.

The paper is clear and easy to follow. The contributions are of strong relevance to the community. I recommend acceptance.

**Additional Comments On Reviewer Discussion:**

Main overlapping concerns revolved around the reliance on GPT-4 as a source of truth, and the ability of JudgeLM to generalize to tasks beyond the training set. For the first concern, authors recognized the limitation and hinted towards future work where multiple diverse teacher models are considered. Authors also referred to tables showing task diversity in the validation set, although the ability of JudgeLM to generalize beyond training tasks remains unclear.

---

### Decision · Program_Chairs · 2025-01-22

Accept (Spotlight)